# Dihydroceramide- and ceramide-profiling provides insights into human cardiometabolic disease etiology

C. Wittenbecher [1,2,3], R. Cuadrat [1,3], L. Johnston [4], F. Eichelmann [1,3], S. Jäger [1,3], O. Kuxhaus[1,3], M. Prada[1,3], F. Del Greco M.[5], A. A. Hicks [5], P. Hoffman [6,7], J. Krumsiek[8], F. B. Hu[2,9] & M. B. Schulze [1,3,10 ✉]

Metabolic alterations precede cardiometabolic disease onset. Here we present ceramide- and dihydroceramide-profiling data from a nested case-cohort (type 2 diabetes [T2D, $n = 775$]; cardiovascular disease [CVD, $n = 551$]; random subcohort [$n = 1137$]) in the prospective EPIC-Potsdam study. We apply the novel *NetCoupler-algorithm* to link a data-driven (dihydro) ceramide network to T2D and CVD risk. Controlling for confounding by other (dihydro) ceramides, ceramides C18:0 and C22:0 and dihydroceramides C20:0 and C22:2 are associated with higher and ceramide C20:0 and dihydroceramide C26:1 with lower T2D risk. Ceramide C16:0 and dihydroceramide C22:2 are associated with higher CVD risk. Genome-wide association studies and Mendelian randomization analyses support a role of ceramide C22:0 in T2D etiology. Our results also suggest that (dh)ceramides partly mediate the putative adverse effect of high red meat consumption and benefits of coffee consumption on T2D risk. Thus, (dihydro)ceramides may play a critical role in linking genetic predisposition and dietary habits to cardiometabolic disease risk.

[1] Department of Molecular Epidemiology, German Institute of Human Nutrition Potsdam-Rehbruecke, Nuthetal, Germany. [2] Department of Nutrition, Harvard T.H. Chan School of Public Health, Boston, MA, USA. [3] German Center for Diabetes Research (DZD), Neuherberg, Germany. [4] Steno Diabetes Center Aarhus, Aarhus, Denmark. [5] Institute for Biomedicine, Eurac Research, Bolzano/Bozen, Italy, affiliated with the University of Lübeck, Lübeck, Germany. [6] Human Genomics Research Group, Department of Biomedicine, University of Basel, Basel, Switzerland. [7] Institute of Human Genetics, University of Bonn, School of Medicine & University Hospital Bonn, Bonn, Germany. [8] Institute for Computational Biomedicine, Englander Institute for Precision Medicine, Department of Physiology and Biophysics, Weill Cornell Medicine, New York, NY, USA. [9] Channing Division of Network Medicine, Department of Medicine, Brigham and Women's Hospital and Harvard Medical School, Boston, MA, USA. [10] Institute of Nutritional Science, University of Potsdam, Potsdam, Germany. ✉email: mschulze@dife.de

Type 2 diabetes (T2D) and cardiovascular disease (CVD) are major worldwide contributors to disease burden and premature mortality[1,2]. Targeted primary cardiometabolic risk prevention requires pathway-specific biomarkers to detect the early metabolic alterations that predispose to developing these common diseases. Pathway-specific biomarkers can help identify at-risk individuals and discover the molecular processes that expose them to higher cardiometabolic risk. Such biomarkers may also help understand the influence of lifestyle on disease risk, enabling precise disease prevention.

Altered blood lipid composition is a common metabolic determinant of T2D and CVD[3]. Among lipids, ceramides are crucial second messengers in systemic signaling cascades, triggering cardiometabolic diseases[4]. In rodents, ceramide metabolites regulate inflammatory signaling, insulin resistance, and cellular stress responses. Genetic modifications of ceramide metabolizing enzymes either protected or predisposed animals for severe metabolic impairments[5,6]. Epidemiological studies have shown associations of ceramides and dihydroceramides with CVD and T2D risk[7–11], suggesting that ceramide-dependent pathogenic mechanisms are also active in human populations.

Concurrently, plasma ceramide concentrations are susceptible to lifestyle modification, including diet. Double-blinded randomized controlled trials (RCTs) have demonstrated that modification of the diet's fatty acid (FA) composition (higher palmitate- vs. linoleic acid-content) alone increased liver fat content and plasma ceramide levels[12,13]. Besides, a *post hoc* analysis of the PREDIMED trial suggested that CVD prevention with a Mediterranean diet intervention particularly alleviated the higher risk of major cardiovascular events in participants with elevated ceramide levels before the intervention[14].

Accordingly, a beneficial composition of the habitual diet was related to lower cardiometabolic disease incidence[15–18]. For example, we and others have shown that red meat and coffee consumption were associated with altered cardiometabolic risk[19–23] and altered lipid metabolism[24–28]. However, the actual metabolic pathways that connect these foods to cardiometabolic risk are still poorly understood. Due to their potential role as disease determinants and the demonstrated sensitivity to dietary exposures, ceramides are plausibly among metabolic mediators of the effect of diet on cardiometabolic risk.

Ceramide metabolism is complex, regulated by over 40 enzymes; these enzymes are subject to multiple regulatory processes and selectively synthesize or degrade groups of ceramides with similar acyl chains[29]. However, it is unclear how molecular pathways in ceramide metabolism are reflected in circulating ceramide profiles. In such situations, data-driven networks can provide information on the biological dependencies that drive the correlation structure of lipidomics profiles[30,31]. We have shown that partial correlation networks of metabolomics data reconstruct molecular pathways[30,31]. Through adjusting for metabolomics network neighbors, our new *NetCoupler-algorithm* controls for confounding by biologically closely related metabolites. Thereby, the robust associations indicate putative direct effects of molecular markers on disease risk and are not attributable to the correlations with other metabolites[32].

Advanced high throughput lipidomics screens generate unprecedented insights into ceramide metabolism[33]. Here we applied the *NetCoupler-algorithm* to ceramide-profiling data from a large human population study to infer the direct effects of specific ceramides and dihydroceramides on the risk of developing T2D and CVD. We then conducted genome-wide association studies (GWAS) on these disease-associated ceramides to learn about inherited biological determinants and select genetic instruments for subsequent Mendelian randomization studies. We also performed hypothesis-generating mediation analyses, estimating the extent to which diet-related (dh)ceramide levels could explain the adverse effects of red meat consumption and the beneficial effects of coffee consumption on T2D risk.

## Results

**Data distribution and the network model.** We used the following short notation for ceramides throughout the manuscript: CerXX:Y for ceramides and dhCerXX:Y for dihydroceramides with XX carbon atoms and Y double-bounds in the acyl chain (Supplementary Table 1). In a pilot study in 35 EPIC-Potsdam participants with two blood samples taken ~6 weeks apart, we assessed the within-person agreement of (dh)ceramide measurements. The intraclass correlation coefficients (ICC) from the pilot indicated fair to excellent reliability of most ceramide—and about half of the dihydroceramide measurements. However, few ceramide measurements and about half of the dihydroceramide measurements showed poor reliability (Supplementary Fig. 1).

The observational analyses were based on the measurement of 12 ceramides and 13 (dh)ceramides from a large lipidomics dataset in two case-cohort samples nested within the prospective EPIC-Potsdam study (775 participants with incident T2D among 1886 at-risk participants, and 551 participants with incident CVD among 1671 at-risk participants). In the random subcohort ($n = 1137$; baseline-prevalent T2D cases excluded), representative for the full EPIC-Potsdam cohort at cardiometabolic risk, the median plasma concentrations ranged between 0.2 nM (Cer18:1) and 42 nM (Cer24:0) for ceramides, and 0.62 nM (dhCer14:0) and 11 nM (dhCer24:1) for dihydroceramides. Median total concentrations (sum of all single compounds within the lipid class) were 91 nM (IQR 76–108 nM) for ceramides and 46 nM (IQR 41–52 nM) for dihydroceramides (Fig. 1A). Log-transformation and z-standardization of the concentrations resulted in similarly scaled, approximately normal distributions (Fig. 1B). Correlation analyses showed moderate to strong correlations between most (dh)ceramides. Partial correlations (conditioning on all other (dh)ceramides) were on average weaker and more specific (Supplementary Fig. 2). Participants with higher total plasma ceramide concentrations and with higher total plasma dihydroceramide ceramide concentration likewise tended to be older, to have a higher waist circumference, to have unhealthy lifestyle habits, and to be on medication (Supplementary Tables 2 and 3). Likewise, participants with incident cardiometabolic diseases had expectedly higher levels of these known risk factors compared to participants who remained cardiometabolic disease-free during follow-up (Supplementary Tables 4 and 5). We adjusted for all these potential confounders in the prospective analyses.

**Dihydroceramide- and ceramide-associated cardiometabolic risk from standard Cox models.** First, we estimated the T2D and CVD risk associated with each single (dh)ceramide without considering the possible influence of other (dh)ceramides. In minimally adjusted models (age and sex only), 9 out of 12 ceramides and 11 out of 13 dihydroceramides were statistically significantly associated with higher T2D risk (FDR < 0.05) (Supplementary Table 6). Further adjustment for lifestyle, anthropometry, medications, blood pressure, and general lipid markers, including total ceramide and dihydroceramide concentration, rendered most of these associations non-significant. However, two ceramides (Cer18:0, Cer22:0) and two dihydroceramides (dhCer20:0, dhCer22:0) remained significantly associated with higher T2D risk and Cer24:0 with lower T2D risk after multiple testing correction (FDR < 0.05). We also observed significant associations of all 12 ceramides and 12 out of 13

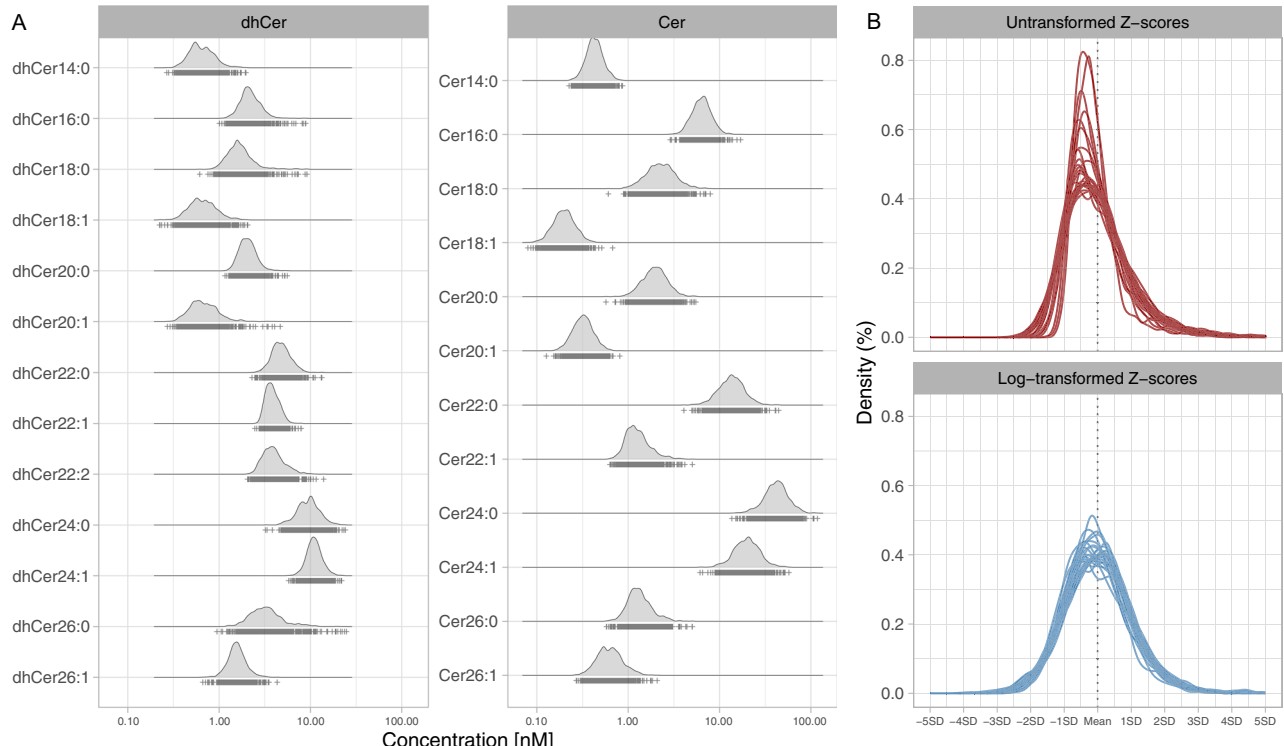

**Fig. 1 Distribution of ceramide and dihydroceramide measurements. A** Distribution of the absolute (dh)ceramide plasma concentrations; note that the *x*-axis is log scaled. **B** Comparison of Z-scores derived from the non-transformed and log-transformed (dh)-ceramide plasma concentrations. Cer ceramide, dhCer dihydroceramide.

dihydroceramides with higher CVD risk in minimally adjusted models. However, in the extensively confounder-adjusted models, only dhCer22:2 was significantly associated with higher CVD risk (FDR < 0.05). Most of the significant CVD associations were rendered non-significant by adjustment for total ceramide and dihydroceramide plasma concentrations (Supplementary Table 6).

**Direct links between the (dh)ceramide network and cardio-metabolic risk.** Ceramide metabolites, depending on their acyl chain, are produced by different enzymes and exhibit distinct signaling functions[34]. Therefore, we were interested in the direct effects of specific (dh)ceramides on cardiometabolic risk, controlling for potentially confounding associations with other, disease-related (dh)ceramides. Our *NetCoupler-algorithm* exploits that adjustment for all network variables is not necessary to block potential confounding and indirect influences in a conditional independence network. Adjustment for a subset of direct network neighbors [i.e., the (dh)ceramides that are directly connected with an edge] is sufficient[32,35,36]. We first learned a graphical representation of the conditional independence structure, the (dh)ceramide network, from lipidomics data in the random EPIC-Potsdam subcohort (Fig. 2). In this data-driven network, most edges reflected known product-substrate-relations in lipid metabolism, such as fatty acid (FA) elongation steps, FA desaturation steps, or desaturation of dihydroceramides to ceramides. Consistent with our previous reports[30,31], the network-encoded conditional independence structure corresponds well with known biological relations.

We used the network to estimate the direct effects of specific (dh)ceramides on cardiometabolic risk, applying Cox proportional hazards regression. To this end, we constructed sets of Cox models for each (dh)ceramide with time-to-disease incidence as the endpoint. All models were extensively adjusted for potential confounders, and the models within each set adjusted for all possible combinations of direct network neighbors of the exposure-(dh)ceramide. We classified (dh)ceramides as having direct effects if they were consistently, statistically significantly (P < 0.05) associated with disease risk across all the network-based adjustment sets.

According to these criteria, three ceramides (Cer18:0, Cer20:0, Cer22:0) and three dihydroceramides (dhCer20:0, dhCer22:2, dhCer26:1) were associated with T2D risk. When simultaneously included in a joint Cox model, including adjustments for the predefined confounder set and total ceramide and dihydroceramide concentration, Cer18:0, Cer22:0, dhCer20:0, and dhCer22:2 were statistically significantly (P < 0.05) associated with higher and Cer20:0 and dhCer26:1 with lower T2D risk (Table 1 and Supplementary Table 7). The three saturated FA (SFA)-containing ceramides were closely related in the network (Fig. 2).

The *NetCoupler-algorithm* also detected associations of Cer16:0 and dhCer22:2 with CVD risk (Supplementary Table 8). In the confounder-adjusted joint model, both (dh)ceramides were statistically significantly (P < 0.05) associated with higher CVD risk (Table 1). In the network, Cer16:0 was linked to the SFA-containing T2D-associated ceramides, while dhCer22:2 was associated with higher risk of both cardiometabolic endpoints (Fig. 2).

In sensitivity analyses, neither additional adjustment of the final model for HDL-cholesterol (Supplementary Table 9) nor exclusion of participants on lipid-lowering medication at baseline (Supplementary Table 10) substantially changed the effect estimates for T2D risk or CVD risk. Similarly, exclusion of participants with disease incidence within the first 2 years of follow-up generated directionally consistent estimates for T2D risk and CVD risk for all selected (dh)ceramides, though the associations of dhCer20:0 with higher and dhCer26:1 with lower T2D risk were substantially attenuated (Supplementary Table 11).

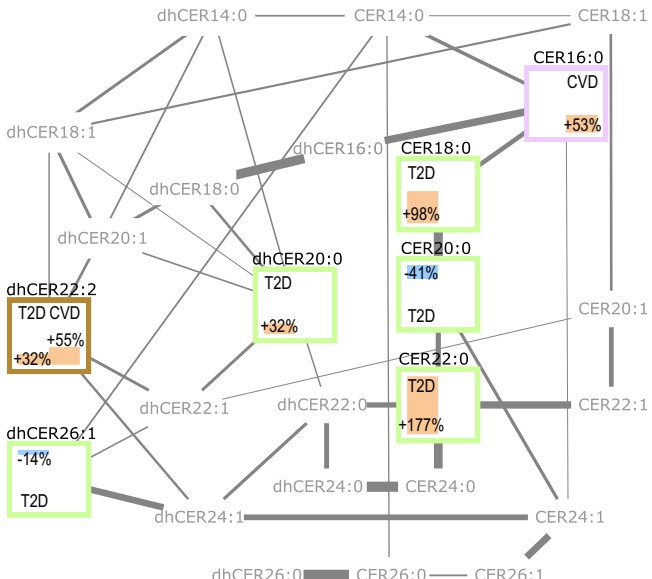

**Fig. 2 Data-driven conditional independence network of (dh)ceramides.**
Bars within nodes show network-adjusted cardiometabolic disease risk.
Left: T2D risk; Right: CVD risk; Orange: increased risk; Blue: decreased risk;
Numbers: percent risk change with 1 standard deviation higher (dh)
ceramide concentration. Frame colors—Green: only T2D-associated;
Purple: only CVD-associated; Brown: T2D- and CVD-associated. CER
ceramide, dhCER dihydroceramide.

| Table 1 Direct links between circulating (dh)ceramides and cardiometabolic risk. | | |
|---|---|---|
| | **T2D** | **CVD** |
| **(dh)ceramide** | **HR (95%CI)** | **HR (95%CI)** |
| Cer16:0 | – | 1.53 (1.15, 2.02) |
| Cer18:0 | 1.98 (1.43, 2.74) | – |
| Cer20:0 | 0.59 (0.39, 0.9) | – |
| Cer22:0 | 2.77 (1.72, 4.47) | – |
| dhCer20:0 | 1.32 (1.08, 1.63) | – |
| dhCer22:2 | 1.32 (1.07, 1.62) | 1.55 (1.23, 1.94) |
| dhCer26:1 | 0.86 (0.74, 0.99) | – |

Hazard ratio (HR) per one standard deviation higher plasma concentration in the EPIC-Potsdam cohort.
Risk estimates are from a model that mutually included all ceramides selected as direct effectors by the *NetCoupler-algorithm* (*see* methods section), further adjusting for total ceramide and total dihydroceramide concentrations, age (strata variable), sex, height, waist circumference, leisure-time physical activity, fasting status, antihypertensive medication, lipid-lowering medication, aspirin, total energy intake, smoking, alcohol consumption, educational attainment, plasma concentrations of triglycerides, total cholesterol, and systolic and diastolic blood pressure; baseline-prevalent T2D cases were excluded from the diabetes risk model, and adjusted for in the CVD risk model.

**Genome-wide association studies on disease-associated (dh)
ceramides.** We conducted a GWAS with the seven disease-related
(dh)ceramide plasma concentrations as the phenotypes in all
participants in the representative EPIC-Potsdam subcohort with
genetic and lipidomics data ($n = 1094$). Then, we looked up SNP-
(dh) ceramide associations at a genome-wide suggestive sig-
nificance level ($p$-value < $10^{-5}$) in independent study populations.
To this end, we used partly unpublished results from a previous
GWAS on ceramides Cer18:0, Cer20:0, and Cer22:0 in the
EUROSPAN consortium[37,38] (Supplementary Data 1), and results
from a GWAS on Cer22:0 in the Framingham Heart Study Off-
spring Cohort published by Cresci et al.[39]. GWAS in these
external cohorts supported the association of SNPs in the *SPTLC3*

gene region with Cer22:0 plasma concentrations (Table 2). Other
suggestive GWAS signals ($p$-value < $10^{-5}$) in EPIC-Potsdam were
either not significant (FDR > 0.05 correcting number of SNPs
available for replication) or not available in the external replica-
tion cohorts and are provided in the supplement (Supplementary
Data 2–8).

**Enrichment of ceramide-associated SNPs in cardiometabolic
disease-related pathways.** Based on all $p$-values from our GWAS
in 1094 EPIC-Potsdam participants, we conducted gene set
enrichment analyses with the GSA-SNP2 software[40]. As the
reference, we considered a curated list of T2D-related pathways
for the T2D-related (dh)ceramides[40], and we generated a curated
list of CVD-related pathways for the CVD-related (dh)ceramides.
We selected enriched gene sets at a $Q$-value of 0.25, a standard
cutoff in gene set enrichment analyses. For T2D, we observed
enriched genetic associations with T2D-associated, long-chain
and very long-chain SFA-containing (dh)ceramides (Cer18:0,
Cer22:0, dhCer20:0, and dhCer26:1) in gene sets related to glu-
cose homeostasis, insulin signaling, and inflammation. For the
very-long-chain FA-containing dhCer22:2, associated with T2D
and CVD risk, enrichment analyses suggested overrepresentation
of genetic associations in gene sets that reflect mitochondrial
dysfunction as well as signaling cascades involved in hemostasis
(Supplementary Fig. 3). No enriched signals in CVD-related gene
sets were detected for the CVD-associated Cer16:0. External data
for replication of the gene set enrichment analyses were not
available.

**Mendelian randomization to evaluate the causal role of cer-
amides.** The association of several SNPs in the *SPTLC3* gene
region with the plasma concentrations of the T2D-associated
Cer22:0 was the single suggestive GWAS signal in EPIC-Potsdam
consistent with the limited available data for external replication.
The detected SNPs in the *SPTLC3* gene region in EPIC-Potsdam
were largely synonymous ($r^2 = 0.96$-1, $D' = 1.0$). The association
of variation in rs680379 with Cer22:0 plasma concentrations had
the lowest $p$-value among SNPs that were available for external
replication in EUROSPAN[37,38] and the Framingham Heart Study
Offspring Cohort[39], and the SNP was also available in a large
GWAS on T2D (DIAGRAM)[41]. Therefore, we used rs680379 as
genetic instrument for a univariable, two-sample Mendelian
randomization study (MR). The results suggested higher T2D risk
in participants with higher genetically predicted Cer22:0 plasma
concentrations. Using the same genetic instrument, we replicated
the MR with the SNP-phenotype association from the two pub-
lished GWAS on plasma ceramides that we used for lookup[37,39]
and found that the MR estimates were also significant (Table 3).
We did not conduct MRs with other (dh)ceramide-endpoint
associations because data for external replication was lacking.

**Ceramides as mediators of putative diet-effects on type 2 dia-
betes.** Habitual intakes of red meat and coffee consumption were
consistently reported as risk factors of T2D[15], but the potential
underlying molecular mechanisms are unclear. A possible
explanation for the relationship with T2D risk is an effect of these
foods on lipid metabolism, possibly involving ceramides. To test
whether association in EPIC-Potsdam were consistent with this
hypothesis, we first assessed if red meat and coffee consumption
were associated with T2D-related (dh)ceramides in a directionally
consistent and statistically significant manner. In mutually
adjusted models and accounting for an extensive set of potential
lifestyle confounders, red meat intake was associated with a
higher concentration of dhCer20:0 and Cer18:0 and lower levels
of Cer20:0 and dhCer26:1 (Fig. 3A). The red meat-related T2D

**Table 2 Genetic variants associated with cardiometabolic disease-related (dh)ceramides.**

| Chr | Rsid | Nearest gene | EPIC-Potsdam beta (SE) | (n = 1094) P | EUROSPAN beta (SE) | (n = 4034) P | FHSOC[a] beta (SE) | (n = 2217) P |
|---|---|---|---|---|---|---|---|---|
| Cer20:0 | | | | | | | | |
| 22 | rs7290187 | GRAMD4 | −0.186 (0.041) | 5.3E−06 | 0.003 (0.001) | 0.011 | NA | NA |
| Cer22:0 | | | | | | | | |
| 20 | rs686548 | SPTLC3 | −0.203 (0.039) | 2.1E−07 | NA | NA | −0.031 (0.005) | 6.8E−11 |
| 20 | rs680379 | SPTLC3 | −0.201 (0.039) | 2.3E−07 | 0.054 (0.007) | 1.3E−13 | −0.031 (0.005) | 4.0E−11 |
| 20 | rs1321940 | SPTLC3 | −0.201 (0.039) | 2.4E−07 | NA | NA | −0.031 (0.005) | 3.5E−11 |
| 20 | rs168622 | SPTLC3 | −0.197 (0.039) | 4.1E−07 | 0.054 (0.007) | 1.2E−13 | −0.031 (0.005) | 3.0E−11 |
| 20 | rs2327451 | SPTLC3 | −0.199 (0.040) | 8.4E−07 | −0.054 (0.008) | 1.4E−12 | −0.029 (0.005) | 5.7E−09 |
| 20 | rs2327452 | SPTLC3 | −0.191 (0.040) | 2.5E−06 | NA | NA | −0.029 (0.005) | 5.6E−09 |
| 20 | rs364585 | SPTLC3 | −0.197 (0.039) | 4.2E−07 | 0.054 (0.007) | 1.1E−13 | −0.031 (0.005) | 3.1E−11 |
| 20 | rs3843765 | SPTLC3 | −0.197 (0.040) | 1.1E−06 | NA | NA | −0.029 (0.005) | 5.7E−09 |
| 20 | rs3848744 | SPTLC3 | −0.190 (0.040) | 3.1E−06 | NA | NA | −0.029 (0.005) | 4.6E−09 |
| 20 | rs3848745 | SPTLC3 | −0.199 (0.040) | 8.4E−07 | NA | NA | −0.029 (0.005) | 4.7E−09 |
| 20 | rs3848746 | SPTLC3 | −0.190 (0.040) | 2.4E−06 | NA | NA | −0.029 (0.005) | 5.5E−09 |
| 20 | rs3903703 | SPTLC3 | −0.198 (0.040) | 8.7E−07 | 0.054 (0.008) | 1.4E−12 | −0.029 (0.005) | 4.8E−09 |
| 20 | rs438568 | SPTLC3 | −0.190 (0.039) | 1.2E−06 | NA | NA | −0.031 (0.005) | 3.5E−11 |
| 20 | rs4508668 | SPTLC3 | −0.199 (0.040) | 8.2E−07 | 0.054 (0.008) | 1.4E−12 | −0.028 (0.005) | 6.0E−09 |
| 20 | rs4814173 | SPTLC3 | −0.198 (0.040) | 9.4E−07 | NA | NA | −0.029 (0.005) | 5.0E−09 |
| 20 | rs4814175 | SPTLC3 | −0.200 (0.039) | 3.2E−07 | NA | NA | −0.031 (0.005) | 2.8E−11 |
| 20 | rs4814176 | SPTLC3 | −0.200 (0.039) | 3.2E−07 | 0.054 (0.007) | 1.0E−13 | −0.031 (0.005) | 3.6E−11 |
| 20 | rs6041735 | SPTLC3 | −0.187 (0.041) | 4.6E−06 | NA | NA | −0.029 (0.005) | 5.1E−09 |
| 2 | rs780094 | GCKR | −0.193 (0.040) | 1.6E−06 | 0.016 (0.007) | 0.022 | NA | NA |
| 2 | rs780093 | GCKR | −0.188 (0.040) | 2.9E−06 | 0.016 (0.007) | 0.022 | NA | NA |

Shown are all associations of SNPs with disease-related (dh)ceramides that were genome-wide suggestively significant ($10^{-5}$) in EPIC-Potsdam and available for replication in EUROSPAN or FHSOC or both. Note that the associations of the SNP associated with Cer20:0 and the two Cer22:0-associated SNPs in the *GCKR*-gene with Cer22:0 were not significant after multiple testing correction (FDR > 0.05); all replication of Cer22:0-associated SNPs in the *SPTLC3* gene region were significant after multiple testing correction (FDR < 0.05) (corrected for the number of SNPs available for replication).

EPIC-Potsdam: 1094 participants, GWAS adjusted for age and sex, ceramides log-transformed, and variance standardized. EUROSPAN[37]: 4034 participants, GWAS adjusted for age and sex, ceramides log-transformed, and variance standardized. FHSOC[39]: 2217 participants, GWAS adjusted for kinship, age, sex, diabetes, smoking status, systolic blood pressure, antihypertensive treatment, body mass index (BMI), and two associated principal components (PC1, PC9) of population stratification. Estimates were reported per standard deviation (EPIC-Potsdam, EUROSPAN) and per µM (FHSOC).

[a]The beta estimates from FHSOC included in the MR were extracted from Table 2 in the publication 'Genetic Architecture of Circulating Very-Long-Chain (C24:0 and C22:0) Ceramide Concentrations' by Cresci et al.[39].

*ITPR1* inositol 1,4,5-trisphosphate receptor type 1 (encoding a plasma membrane receptor), *GRAMD4* GRAM domain containing 4 (innate immune response), *SPTLC3* Serine Palmitoyltransferase Long Chain Base Subunit 3 (encoding the key enzyme in sphingolipid biosynthesis), *GCKR* Glucokinase Regulator (encoding the glucokinase regulator protein), PM plasma membrane SL sphingolipid, FHSOC Framingham Heart Study Offspring Cohort.

risk in EPIC-Potsdam (HR per 2 SD higher intake 1.31, 95%CI 1.01–1.71) was largely attenuated by adjustment for the red meat-associated ceramides (proportion explainable 62%, 95%CI 9% to 100%) (Fig. 3B). Coffee consumption was associated with lower concentrations of the high-risk dihydroceramide C22:2 (Fig. 3C). Adjusting the inverse coffee-T2D association (HR per 2 cups 0.87, 95%CI 0.78–0.98) for dhCer22:2 attenuated the inverse association of coffee with T2D risk by 43% (95%CI 10% to 99%) (Fig. 3D). Thus, our mediation analyses results are consistent with the hypothesis that divergent effects on ceramide metabolism partly mediate the opposite putative effects of red meat and coffee consumption on T2D risk.

## Discussion

In this prospective study in a baseline-healthy, free-living population, a metabolic network based on deep ceramide and dihydroceramide-profiling data revealed several associations of specific (dh)ceramides with cardiometabolic disease risk robust against adjustment for other (dh)ceramides. When simultaneously included in a confounder- and total ceramide and dihydroceramide-adjusted Cox model, high plasma concentrations of Cer18:0, Cer22:0, dhCer20:0, and dhCer22:2 were associated with a higher T2D risk, while Cer20:0 and dhCer26:1 were associated with lower T2D risk. The high T2D risk associated with Cer18:0 and Cer22:0 suggests that these compounds may be directly involved in molecular mechanisms that implicate ceramide metabolism in T2D etiology. Mendelian randomization estimates were consistent with an effect of Cer22:0 on T2D risk, and gene set enrichment analyses suggestively linked Cer18:0 to insulin signaling and both ceramides to cytokine-induced inflammation. Mediation analyses suggested differential influences of high red meat and high coffee consumption on ceramide metabolism, potentially explaining the putative opposite effects of the two foods on T2D risk. Moreover, when simultaneously included into the same confounder- and total ceramide and dihydroceramide-adjusted model, Cer16:0 and dhCer22:2 were both associated with higher CVD risk. Enrichment analyses suggested enrichment of dhCer22:2-associated SNPs in gene sets related to the regulation of hemostasis and platelet aggregation.

Prospective human studies showed an association of ceramides with T2D risk and diabetes-related traits. In the Strong Heart Study, Cer16:0, Cer18:0, Cer20:0, and Cer22:0 were associated with insulin resistance[8]. The FINRISK-cohort reported that the Cer18:0-to-Cer16:0-ratio was associated with higher T2D risk[42], suggesting the relation of Cer18:0 to shorter chain precursors as a predictor for T2D incidence. Another study associated (dh)ceramides with T2D incidence in mice and humans, particularly those with 18 and 22 carbon atoms in the acyl chain[43]. Despite heterogeneity due to different included (dh)ceramides and diverse modeling approaches, these observations are generally consistent with our results. Based on our network adjustments, we linked saturated LCFA-containing (dh)ceramides to T2D risk in a chain

---

**Table 3 Univariable, two-sample Mendelian randomization studies using genetic proxies to estimate effects of Cer22:0 on the risk of T2D.**

**T2D-associations from DIAGRAM ($n = 74{,}124$ T2D cases and 824,006 controls)**

| Cer22:0-associations from | EPIC-POTSDAM ($n = 1094$) rs680379 | EUROSPAN ($n = 4034$) rs680379 | FHSOC[a] ($n = 2217$) rs680379 |
|---|---|---|---|
| WR | 0.070 | 0.259 | 0.452 |
| SE | 0.032 | 0.120 | 0.210 |
| *p*-value | 0.031 | 0.031 | 0.031 |

Several SNPs near the *SPTLC3* gene were associated with Cer22:0 plasma concentrations in GWAS in the EPIC-Potsdam study. We compared our SNP-phenotype associations with data from two published GWAS on Cer22:0[37,39], and SNP-T2D associations were drawn from DIAGRAM (T2D)[41]. Among the SNPs available in all these studies, the rs680379 association with Cer22:0 plasma concentrations in EPIC-Potsdam had the lowest *p*-value ($P = 2.3\text{E-}07$), and rs680379 was therefore used for a univariable, two-sample MR in EPIC-Potsdam. We replicated the MR with SNP-Cer22:0 association from the independent cohorts (EUROSPAN and FHSOC). SNP-ceramide 22:0 associations were reported per standard deviation (EPIC-Potsdam, EUROSPAN) and per µM (FHSOC).
[a]The beta estimates for the association of rs680379 with Cer22:0 in FHSOC included in the MR were extracted from Table 2 in 'Genetic Architecture of Circulating Very-Long-Chain (C24:0 and C22:0) Ceramide Concentrations' by Cresci et al.[39].
WR Wald ratio, SE standard error, T2D type 2 diabetes, CAD Coronary artery diseases, MR Mendelian randomization, FHSOC Framingham Heart Study Offspring Cohort.

---

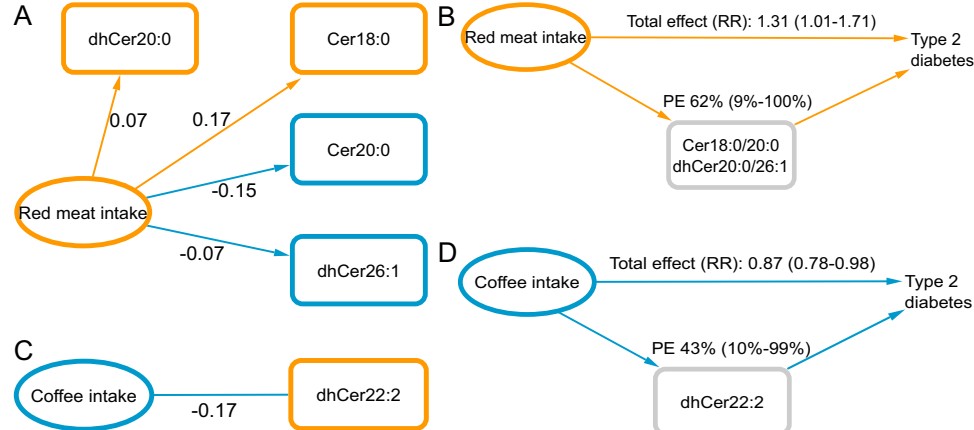

**Fig. 3 Mediation analysis.** **A** Adjusted effect estimates (beta coefficients) of red meat on T2D-related (dh)ceramides (direction of associations consistent with mediation hypothesis; p-values < 0.05, one-sided t-test). **B** Attenuation of the putative effect of red meat on T2D risk after adjustment for red meat- and T2D-related (dh)ceramides. **C** Adjusted effect estimate (beta coefficient) of coffee on T2D-related dhCer22:2 (direction of the association consistent with the mediation hypothesis; p-value < 0.05, one-sided t-test). **D** Attenuation of the putative effect of coffee on T2D risk after adjustment for coffee- and T2D-related dhCer22:2. All models were extensively adjusted for potential confounders (age, sex, fasting status, total energy intake, leisure-time physical activity, medication, smoking, alcohol consumption, and education). Blue indicates inverse association (i.e., lower ceramide concentration or T2D risk), orange: positive association (i.e., higher ceramide concentration or T2D risk). Total effect is the confounder-adjusted hazard ratio (95% CI) per exposure unit: red meat, 2 SD (~1 portion per day); coffee, two cups (300 mL) per day. PE Proportion explainable, i.e., relative attenuation of the total effect through mediator-adjustment. Cer ceramide, dhCer dihydroceramide.

length-dependent manner and additionally detected risk markers among VLCFA-containing (dh)ceramides. In a mutually adjusted model, high levels of Cer18:0, Cer22:0, dhCer20:0, and dhCer22:2 were associated with higher T2D risk, while Cer20:0 and dhCer26:1 were associated with lower risk.

In cells, ceramide signaling orchestrates the metabolic response to elevated levels of non-esterified FAs[4]. To this end, ceramides induce triglyceride synthesis (for example, by translocation of CD36 to the plasma membrane and by induction of SREBP genes[44–46]), downregulate nutrient supply (among others by insulin desensitization and downregulation of lipolysis[47–52]), and stimulate FA-utilization (e.g., by decreasing mitochondrial efficiency, which diminishes feedback inhibition of beta-oxidation[53,54]). Under a prolonged metabolic challenge, ceramides also link cellular stress to immune responses, apoptosis[55,56], and fibrosis[57,58]. Thereby, intracellular concentrations of LCFA-containing ceramides serve as nutrient sensors. Accordingly, genetic knockout of Cer18:0-producing ceramide-synthase (CerS)-1 protected mice from the detrimental effects of a high-fat diet on systemic glucose homeostasis[6]. Our study consistently linked Cer18:0 to strongly elevated T2D risk, while its direct network-neighbor Cer20:0 was moderately inversely related to T2D risk when simultaneously included in the same Cox model. Our genetic analyses consistently suggested functions of LCFA-containing ceramides in metabolic regulation, particularly linking them to insulin-signaling pathways. Our results specifically suggest that the interference of Cer18:0 with insulin sensitivity, which was demonstrated in animal models, is linked to T2D development in a free-living human population.

Studies in rodents and humans demonstrated that ceramide signaling partly mediates the adverse effects of an unfavorable dietary FA composition on metabolic health[12,13,45]. We related high habitual red meat consumption to an adverse saturated LCFA-signature in ceramides, specifically higher levels of Cer18:0. Moreover, we observed a marked effect attenuation by controlling the red meat-related T2D risk for LCFA-containing ceramides, suggesting that the higher T2D incidence among people with high red meat consumption is partly explainable by red meat-induced alteration of the saturated LCFA-composition of ceramides.

The quantitatively most abundant ceramide synthase in the human liver is CerS2, which synthesizes Cer22:0. Genetic ablation of CerS2 in mice suppresses the hepatic adaptation to nutritional challenges. CerS2-knockout mice were protected against liver fat accumulation and elevated blood sugar in overfeeding regimens but developed severe hepatic pathologies[59,60]. However, hepatocytes of CerS2-knockout mice were protected against lipid-induced *TNF-α/NF-κB*-dependent inflammation and apoptosis[61]. In our study, Cer22:0 was among the quantitatively most abundant ceramides in plasma, and it was the strongest T2D risk marker. Our GWAS results further linked Cer22:0 to *NF-κB* activation, and Mendelian randomization suggested that it might play a biological role in T2D development. Our results suggest that the human plasma concentration of Cer22:0 may serve as a biomarker for metabolically induced cellular stress and inflammatory signaling that predispose to T2D.

We also observed associations of dhCer22:2 with higher and dhCer26:1 with lower T2D risk. Gene set enrichment analysis suggested enrichment of dhCer22:2-associated SNPs in mitochondrial function-related pathways and dhCer26:1-associated SNPs in insulin signaling- and inflammation-related pathways. Other studies also linked dihydroceramides with 22 carbon atoms acyl chains to insulin sensitivity and hepatic inflammation but did not assess dhCer22:2 concentrations[43,62]. Our network-adjusted analyses suggested dhCer22:2 and dhCer26:1 as new independent T2D risk markers and warrant external validation.

Among dietary factors, coffee was associated with lower cardiometabolic risk[19–21], and the effect of coffee on hepatic lipid metabolism is a potential explanation. Animal studies demonstrated that coffee and its components affect critical regulators of lipid metabolism, including SREBP1, CD36, and PPARα and PPARγ[25–28], affecting lipid uptake, excretion, and FA-metabolism in the liver. As discussed above, ceramides connect nutrient sensing to the regulation of cellular stress responses; and our gene set enrichment analysis suggested that dhCer22:2 may reflect metabolic stress signals and mitochondrial dysfunction. We observed lower concentrations of dhCer22:2 associated with coffee consumption and adjusting for this biomarker substantially attenuated the inverse association of coffee consumption and T2D risk. These observations are consistent with the hypothesis that modification of ceramide metabolism could partially explain the beneficial effects of coffee on cardiometabolic health.

Several studies showed that plasma ceramide concentrations predict CVD risk[7,9,10,14,63,64]. Besides distinct source populations, the different coverage of (dh)ceramides and different modeling strategies complicates the comparison of these studies. We detected Cer16:0 and dhCer22:2 as independent CVD risk markers, using comprehensive lipidomics profiles and a modeling strategy targeting risk association robust against adjustment for the total ceramide and dihydroceramide concentrations and other (dh)ceramides.

Previous reports of Cer16:0 and Cer18:0-associations with higher CVD risk[14,63,64] are consistent with our confounder-adjusted single (dh) ceramide models that did not adjust for network neighbors and total dihydroceramide and ceramide concentrations. However, in our study, only the association of Cer16:0 with CVD risk was robust against adjustment for total ceramide and dihydroceramide concentrations and network neighbors.

We found a robust association of dhCer22:2 with CVD risk in EPIC-Potsdam and did not identify previous reports of the CVD risk association. The gene set enrichment suggested possible involvement in immune response, platelet aggregation, and cell–cell interaction involved in hemostasis. Experimental studies demonstrated that VLCFA-containing ceramides link inflammatory signals to vascular pathologies[65,66]. Genetic and pharmacological inhibition of type 2-neutral sphingomyelinase in mice reduced the circulating VLCFA-containing ceramide concentrations and prevented lipid-induced atherosclerosis[67]. Consistently, the platelet-activating factor activates ceramide production in erythrocytes, leading to their adhesion[68]. Moreover, VLCFA-containing ceramides are functionally involved in necroptosis[69], providing a biological link to vascular health and cardiac cell death. Our genetics and observational findings suggest dhCer22:2 may be a biomarker at the interface of lipid metabolism, inflammatory signaling, and cardiovascular health.

Substantial evidence from animal models supports a causal role of specific (dh)ceramides in cardiometabolic disease development[5,6,44,70,71]. Human intervention studies demonstrated an impact of diet composition on ceramide metabolism[12,13]. Against this background, our results suggest that (dh)ceramide profiling in intervention studies may help to understand the molecular underpinnings of the effect of dietary composition on cardiometabolic health. Plasma ceramide profiling may provide pathway-specific cardiometabolic risk markers, with LCFA-ceramides potentially reflecting metabolic impairment[5,6] and VLCFA-containing ceramides potentially reflecting immune responses and cell–cell interactions[65–69]. However, our study also suggests that the specificity of (dh)ceramides as molecular pathway markers depends on simultaneous assessment and modeling of a comprehensive (dh)ceramide profile.

Our study had limitations. Although delivering a very comprehensive lipidomics screen, the manufacturer (Metabolon®) did

not disclose indicators of the technical variance for single lipids. We partly compensated for this lack of transparency by assessing the intra-individual variance of the single lipid measurements over several weeks in a pilot study, assessing the temporal stability of the (dh)ceramide measurements. Some (dh)ceramides showed substantial within-person variance over several weeks. Under the assumption that the introduced variance is unrelated to the disease risk, poor reliability is expected to bias single measurement-based risk estimates towards the null. Accordingly, most disease-associated (dh)ceramides had fair to excellent ICCs in our reliability study.

In addition, observed associations might be attributable to unmeasured confounding. In combination with experimental data, our selection of specific chain-length (dh)ceramides with direct effects on disease risk can be useful to elucidate molecular mechanisms. However, for other applications, including risk prediction, the total effect of a biomarker is more critical, and it might not be advantageous to adjust for other correlated (dh)ceramides or total levels of ceramides and dihydroceramides.

The p-value-based variable selection and inferences in our observational and genetic analyses depended on the sample size, complicating comparison to studies with different statistical power. Particularly, the GWAS on ceramide risk markers had limited statistical power. External GWAS data to validate SNP-lipid associations in independent cohorts was not available for most ceramides and all dihydroceramides. The pathway enrichment analysis generated plausible biological insights from genetic associations with less stringent significance cutoffs, but datasets for replicating our findings were not available.

The limited statistical power of the GWAS may also partly account for only detecting one reliable instrument for the MR study of Cer22:0 on T2D risk, using a single SNP from the SPTLC3 gene region, which impeded checks for horizontal pleiotropy. Genetic variants in this gene region were also implicated in other lipid and metabolic traits. However, the SNPs were linked to a gene that encodes a subunit of a key enzyme in sphingolipid biosynthesis, and the MR results were replicated with SNP-phenotype associations from independent cohorts. Still, the IVs' pleiotropic effects on other ceramides must be assumed, and the attribution of the effect to Cer22:0 depends on the validity of our network-adjusted observational analysis. Therefore, the MR estimates alone do not provide conclusive evidence on causality but complement the observational estimates due to distinct sources of bias. Our findings encourage larger GWAS on ceramides that may also generate more genetic instruments for MR studies. Mediation analyses in observational data do not prove causality but generate testable hypotheses, which warrant validation in controlled trials.

To conclude, our study indicates that the cardiometabolic risk associated with (dh)ceramide plasma concentrations depends on the contained acyl chain, especially if models are conditioned on other disease-related (dh)ceramides and total ceramide and dihydroceramide concentrations. These observations are consistent with the hypothesis that specific (dh)ceramides are involved in distinct molecular mechanisms of cardiometabolic disease etiology, which coincides with evidence from animal models. Our genetic analyses also suggested the implication of the disease-related (dh)ceramides in cardiometabolic disease-related molecular pathways. Furthermore, we showed that adjustment for a few T2D-related (dh)ceramides markedly attenuated the adverse effect of red meat and the protective effect of coffee consumption on T2D risk, consistent with the hypothesis that their effect on ceramide metabolism partially mediates the effect of these foods on T2D risk. Altogether, these results indicate that circulating (dh)ceramide profiles integrate information on the exposure to genetic and environmental cardiometabolic risk

factors and may be applied as pathway-specific biomarkers for cardiometabolic health.

## Methods

All EPIC-Potsdam participants gave informed consent for biomedical research use of their data, and the study was approved by the Ethics Committee of the State of Brandenburg, Germany[72]. The study participants did not receive monetary compensation. All work was performed in accordance with the Declaration of Helsinki.

### Study population

*EPIC-Potsdam.* The prospective EPIC-Potsdam cohort study includes 27,548 participants (16,644 women and 10,904 men) recruited within an age range of 35–65 years from the general population between 1994 and 1998[72]. Participants were then actively contacted by sending out questionnaires and, if necessary, by telephone every 2–3 years, with response rates between 90% and 96% per follow-up round[73].

Nested case-cohorts were constructed for efficient studies into molecular phenotypes and disease risk. The case-cohort design relies on a randomly drawn subsample (the subcohort) and oversampling of all incident disease cases in the full cohort during the study period to boost the statistical power. Statistically accounting for the oversampling of cases, this design provides unbiased risk estimates for the full cohort[74]. The subcohort (n = 1137; baseline-prevalent T2D cases excluded) was drawn from all participants who provided blood at baseline (n = 26,437). Additionally, for each endpoint, all incident cases in the full cohort until a specified censoring date were included (CVD: 551 incident cases, 28 in the subcohort; T2D: 775 cases, 26 in the random subcohort).

For T2D, the censoring date was the 31st of August 2005 (820 incident cases). After excluding participants with missing follow-up information, prevalent diabetes at recruitment, insufficient blood specimens, or non-verifiable information on diabetes incidence, the analytical sample comprised 1886 participants (1000 women and 886 men), including 775 participants with incident T2D from whom 26 were part of the subcohort. The median follow-up time for T2D was 6.5 years (interquartile range 6.0–8.7 years).

For CVD, the censoring date was the 30th of November 2006, with 583 incident primary cardiovascular events occurring during the study. After equivalent exclusions (using prevalent and non-verifiable CVD instead of diabetes as exclusion criterion), the CVD sample comprised 1671 participants (892 women and 779 men), including 551 participants with incident CVD (283 only myocardial infarction, 257 only strokes, 11 both) from whom 28 were part of the subcohort. The median follow-up time for CVD was 8.4 years (interquartile range 7.6–9.2 years).

*Baseline assessment.* The baseline examination included anthropometric and blood pressure measurements, a personal interview and a questionnaire on prevalent diseases and sociodemographic and lifestyle characteristics (including physical activity, education, and medication), and a validated semi-quantitative food frequency questionnaire (FFQ). Among other foods, the habitual intake of unprocessed and processed red meats and coffee was assessed[75,76]. We defined total red meat as the sum of unprocessed red meat and processed meat. The correlations between quantitative repeated assessment of red meat, processed meat, and coffee consumption were 0.73, 0.77, and 0.70 from FFQs 6 months apart, indicating good to excellent reproducibility[77]. Anthropometric measurements and physical examinations were conducted by trained medical personnel. BMI was calculated as body weight in kilograms divided by squared height in meters. Waist circumference was measured midway between the lower rib margin and the superior anterior iliac spine to the nearest 0.5 cm[78,79]. Blood pressure was measured in a standardized procedure with oscillometric devices (BOSO-Oscillomat, Bosch & Sohn, Jungingen, Germany), and the mean of second and third reading was used[80].

At baseline, blood samples were drawn under standardized conditions regarding room temperature according to the study protocol and stored in liquid nitrogen (−196 °C) or deep freezers (−80 °C). Per participant, 30 ml of blood were collected, of which 20 ml were filled in Monovettes containing citrate. Samples were separated in serum, plasma, buffy coat, and erythrocytes and aliquoted into 0.5 ml straws as previously described in detail[81].

### Laboratory measurements. 
For all laboratory measurements, samples were randomly distributed across batches independent of case status, and all laboratory and data-processing steps were performed blind to the case status.

*Lipid profiling.* The (dh)ceramide-profiling data was generated with Metabolon (Morrisville, US) using the Metabolon® Complex Lipid Panel. The platform generates the molecular species concentration and complete fatty acid composition of each covered lipid class, including 13 dihydroceramides and 12 ceramides. From plasma samples, lipids were extracted in methanol:dichloromethane, concentrated under nitrogen, and reconstituted in ammonium acetate dichloromethane:methanol (50:50). The extracts were directly infused into the ionization source of a Sciex SelexION® −5500 QTRAP mass spectrometer. After ionization, the lipids passed through SelexIon differential mobility spectrometry (DMS), in which voltages are applied that selectively allow the passage of only a specific lipid

class at any given time. After the DMS filtering, lipids entered the Multiple Reaction Monitoring (MRM), where the lipid mass and its characteristic fragment were measured. The Metabolon® Complex Lipid Panel included >50 isotopically labeled internal standards introduced in the biological sample early in the process and permitted accurate quantitation of lipids across and within classes. According to Metabolon®, the coefficients of variation (CVs) of lipid class concentrations are all below 10% and the median CV of species at a 1uM concentration in serum or plasma is ~5%. In a preceding analysis, we estimated intraclass correlation coefficients (ICCs) of repeated blood samples taken several weeks apart. The ICC relates intraindividual to between-person variation, indicating biological stability of the measurements, and we used Rosner's classification of ICCs (ICC < 0.40 poor reproducibility; ICC from 0.40–0.75 fair to good reproducibility; ICC > 0.75 excellent reproducibility)[82].

*Genetics.* We only considered the random subcohort participants for the genetic analyses, excluding prevalent T2D and CVD cases ($n = 1094$). The DNA was extracted from buffy coats using the chemagic DNA Buffy Coat Kit special on a Chemagic Magnetic Separation Module I (PerkinElmer Chemagen technologies, Baesweiler, Germany) according to the manufacturer's instructions. Eligible samples were genotyped with three different genotyping arrays as part of different larger genotyping projects: Human660W-Quad_v1_A ($n = 328$), HumanCore Exome-12v1-0_B ($n = 587$) and Illumina InfiniumOmniExpressExome-8v1-3_A DNA Analysis BeadChip ($n = 179$). Genotyping and quality control of the Human660W-Quad_v1_A and HumanCoreExome-12v1-0_B chips were described elsewhere[83]. Genotyping using the Illumina InfiniumOmniExpressExome-8v1-3_A DNA Analysis BeadChip was performed in the Life and Brain Center in Bonn, Germany. This array contains about 960 000 genetic variants, allowing to genotype 77% of all common genetic variants within the human genome. Additionally, a 250 K high-value exome content, discovered through exome sequencing studies, is covered by the chip. The DNA was processed according to the manufacturer's instruction using an automatized, LIMS controlled workflow, and the arrays were finally scanned using an Illumina iScan bead arrays reader. Genotype calling and quality control of the samples were carried out jointly in all 1094 samples using Illumina's GenomeStudio v2011.1 software suite. Protocols suggested by the CHARGE consortium[84], Anderson et al.[85]., and Guo et al.[86]. were used to derive the final dataset. zCall with a threshold of seven was applied[87] to improve the genotype calling for rare variants. Samples with low call rate, discordant sex information (F-value between 0.2 and 0.8), related or duplicated individuals (IBD > 0.185), individuals with divergent ancestry, or unclear sample allocation were excluded from further analysis ($n$ after exclusions = 1094). Phasing and imputation were conducted using the Michigan Imputation Service[88]. The Haplotype Reference Consortium (release 1.1) was used as a reference panel[89]. Before imputation, pre-phasing was applied using Eagle2[90,91]. Imputation was carried out in four separated datasets (one for each genotyping chip or two for the HumanCoreExome-12v1-0_B chip) using minimac3[88]. Pre- and post-imputation tools (HRC-1000G-check-bim.v4.2.9, icv.1.0.5) for checking data quality were applied[92]. The four imputed files were merged using bcftools[93], keeping the four merged files' minimal $R^2$ score. After, the SNPs were filtered by $R^2$, keeping those with values >0.6. Data were available for the 22 autosomes but not for the sex chromosomes.

*Targeted biomarkers.* The automatic ADVIA 1650 analyzer (Siemens Healthcare, Erlangen, Germany) was used to assess plasma levels of total cholesterol and triglycerides, and we applied a sex-specific correction for dilution with citrate (correction factor 1.16 for women and 1.17 for men)[94].

### Case ascertainment

*T2D.* Systematic information sources for the incidence of T2D were self-report of diagnosis, T2D-relevant medication, or dietary treatment due to T2D diagnosis during follow-up. Additionally, death certificates and information from tumor centers, physicians, or clinics that provided assessments for other diagnoses were screened for an indication of incident T2D. For participants classified as potential cases based on that information, a standard inquiry form was sent to the treating physician. Only physician-verified cases diagnosed with T2D [International Statistical Classification of Diseases and Related Health Problems (ICD)-10 code: E11] and a diagnosis date after the baseline examination were considered confirmed incident cases of T2D.

*CVD.* Incident CVD was defined as the incidence of non-fatal and fatal myocardial infarction (MI) and stroke (ICD-10 codes: I21 for acute MI, I63.0 to I63.9 for ischemic stroke, I61.0 to I61.9 for intracerebral and I60.0 to I60.9 for subarachnoid hemorrhage, and I64.0 to I64.9 for unspecified stroke). The incidence of CVD was assessed by participant self-report or based on information from death certificates. Self-reported incidence was then validated by contacting the treating physicians, including assessing the ICD-10 code, date of occurrence, and further information on symptoms and diagnostic criteria used in the WHO MONICA study. For myocardial infarction, diagnostic criteria included clinical symptoms, electrocardiograms, heart enzymes, and known coronary heart disease. The stroke diagnosis was based on anamnesis, clinical symptoms, CT/MRT, angiogram, lumbar puncture, echocardiogram, Doppler, and ECG, plus imaging techniques if available.

Participants with silent cardiovascular events that have not been documented within 28 days after occurrence were excluded as non-verifiable cases from all analyses.

### Statistics

*Data preparation.* A moderate fraction of covariable information was missing (waist circumference, $n_{missing} = 2$; BMI, $n_{missing} = 12$; blood lipids, $n_{missing} = 82$; blood pressure, $n_{missing} = 148$). Single imputation was used to impute these missing values, applying the "predictive mean matching method" from the SAS procedure PROC MI. The "predictive mean matching method" draws information from other covariables to predict missing values and, compared with linear regression, generally generates more plausible imputed variable distributions[95]. The following variables contributed to the prediction of missing values: incident case (T2D/CVD) during follow-up (yes, no), sex, age, height, smoking, leisure-time physical activity (sports, biking, gardening), drug treatment (antihypertensive, lipid-lowering, aspirin), prevalent disease status (T2D, CVD), total energy intake, intakes of whole-grain bread, grain flakes, grains, and muesli, fresh fruit, raw vegetables, cooked vegetables, nuts, coffee, high-energy soft drinks, fish, red meat and processed meat, total alcohol consumption, and educational attainment.

Smoking was modeled in four categories (never smoker, ex-smoker, current smoker < 20 units/day, current smoker ≥ 20 units/day). Alcohol intake was modeled in six sex-specific intake level categories. The alcohol consumption-categories in men were: abstainers, 0–6 g/d, >6–12 g/d, >12–24 g/d, >24–60 g/d, >60–96 g/d, >96 g/d. The alcohol consumption-categories in women were: abstainers, 0–6 g/d, >6–12 g/d, >12–24 g/d, >24–60 g/d, >60 g/d. Coffee intake was modeled as cups (150 mL) per day and meat intake as grams per day. Educational attainment was modeled in three categories (in or no vocational training/vocational training, technical college degree, university degree). Leisure-time physical activity was modeled as average weekly hours. Fasting status was modeled as a binary variable (≥8 h, yes/no).

The few participants with missing (dh)ceramide values (three for dhCer14:0, 5 for dhCer18:1 and 20:1 each, 13 participants in total) were excluded from all analyses that included these variables. The (dh)ceramide concentrations tended to be right-tailed. Therefore, we log-transformed (dh)ceramide concentrations, which resulted in approximately normal distributions, and z-scaled the log-transformed values. Accordingly, all regression estimates were reported per 1 SD.

*Prentice-weighted Cox models for (dh)ceramide-cardiometabolic risk analyses in the case-cohort.* Associations between (dh)ceramides and disease risk were evaluated in Cox proportional hazards regression models with age as the underlying time scale. Study exit was determined by a diagnosis of diabetes or CVD, dropout, or censoring time, whichever came first. The case-cohort design was accounted for by Prentice weighting[96].

*The NetCoupler-algorithm.* We aimed to estimate the direct effects of (dh)ceramides on cardiometabolic disease risk that could not be attributed to the influence of related ceramide metabolites. Therefore, we developed a graphical model-based method, the *NetCoupler-algorithm*[32]. In a first step, we estimated a network model of conditional dependencies, where edges represent covariance between two (dh)ceramides that could not be explained by adjustment for any subset of other (dh)ceramides. To this end, we applied an order-independent implementation of the causal structure learning PC-algorithm[36,97]. The resulting network graphically encoded the family of causal models that could have generated the observed conditional independence structure, i.e., the skeleton of the data-generating DAG. This conditional independence network was then used to detect links between individual metabolites and disease incidence that could not be explained by confounding influences through other (dh)ceramides. By definition, at least one subset of direct neighbors is sufficient to block confounding from the whole network[35]. However, sufficient adjustment sets could not be unambiguously read from the graph because the edges were not directed. Therefore, the *NetCoupler-algorithm* iterates for each metabolite through adjustment for all possible combinations of direct network neighbors. A metabolite is then only classified as a direct effector if the association with disease incidence is robust across all these sub-models (Supplementary Fig. 4). The analyses were conducted with a developmental implementation (available upon request). We provide detailed documentation and ready-to-use software implementation of the *NetCoupler-algorithm* in the R statistical programing language on GitHub (https://github.com/NetCoupler).

An edge between any possible pair of ceramides was detected based on dependency at an alpha level of 0.05, conditioning on any subset of other ceramides to learn the network. To evaluate the direct link of each ceramide with disease incidence, iterative Cox models were used. Thereby, each ceramide was associated with time-to-disease-incidence, adjusting for all possible combinations of direct neighbors in the ceramide network. All models were additionally adjusted for total ceramide and total dihydroceramide concentrations, age in years (strata variable), sex, height, waist circumference, leisure-time physical activity, fasting status, antihypertensive medication, lipid-lowering medication, aspirin, total energy intake, smoking, alcohol consumption, education, plasma concentrations of triglycerides, total cholesterol, and systolic and diastolic blood pressure; baseline-prevalent T2D cases were excluded from the diabetes risk model, and adjusted for in the CVD risk model. Ceramides that were directionally consistent and

statistically significantly (alpha < 0.05) associated with the disease endpoint across all neighbor-adjusted models were classified as direct effects. Because they can only be confounders but not mediators, each newly identified direct effector ceramide was included in the fixed adjustment set, and the procedure was repeated until no further direct effects were detected. Finally, for each endpoint, all the selected directly disease-associated (dh)ceramides were simultaneously included into the same Cox model, adjusted for the full set of above-defined covariables, rendering the mutually adjusted disease hazard ratios.

*Genome-wide association study.* The software QCtool v1.4 was used to filter the SNPs by SNP missing rate (removed ≥ 0.05), minimum allele frequency (MAF) (removed out of interval [0.05–0.5]), and Hardy–Weinberg equilibrium (removed −log10(p-value) ≥ 3). Then, we used SNPtest v2.5.2 for exploratory single variant association analysis (n ~ 5,339,213 markers) as exposures and the log-transformed and z-standardized (dh)ceramides as an outcome. We considered p-values below $10^{-5}$ suggestively significant. We assumed a frequentist additive genetic model (method expected: genotype dosage), adjusted for age at recruitment and sex. Variants were mapped to Ensembl annotation version 84 (GRCh37)[98], and we used the Ensembl Variant Effect Predictor for annotation[99].

For the GWAS on Cer18:0, Cer20:0, and Cer22:0, we performed lookup studies with partly unpublished results from EUROSPAN (European special populations research network: quantifying and harnessing genetic variation for gene discovery, n = 4034), a consortium involving five European populations focusing on the genomics of >300 phenotypes including lipidomics, that were measured at the Institute for Clinical Chemistry and Laboratory Medicine, Regensburg University Medical Center (Germany), using electrospray ionization tandem mass spectrometry (ESI-MS/MS) in positive ion mode. Genetic association tests between lipid and allele dosage were performed using a mixed model approach implemented with the 'mmscore' option in the GenABEL software. Results from the five populations were combined using inverse variance weighted fixed-effects model meta-analyses using the METAL software. The other (dh)ceramides associated with cardiometabolic risk in EPIC-Potsdam were not available in EUROSPAN. We also compared our suggestively significant GWAS results on Cer22:0 in EPIC-Potsdam with published SNP-Cer22:0 associations from the Framingham Heart Study Offspring Cohort (n = 2217). To this end, we extracted beta estimates and p-values from Table 2 in the publication by Cresci et al.[39]. The other (dh)ceramides associated with cardiometabolic risk in EPIC-Potsdam were not available in the Framingham Heart Study Offspring cohort.

We used summary-level data for the association of ceramide-associated SNPs with T2D obtained from the DIAbetes Genetics Replication And Meta-analysis (DIAGRAM) consortium, including 32 studies with a total of 898,130 individuals (74,124 with T2D and 824,006 without) of European ancestry[41]. In that resource, the Haplotype Reference Consortium reference panel was used for all component studies except deCODE GWAS, which was imputed using a population-specific reference panel (30,440 Icelandic haplotypes)[41]. We used the T2D data without BMI adjustment. The EPIC-Potsdam GWAS data were included in the EPIC-Interact Consortium, which contributed GWAS data to the used DIAGRAM meta-analysis. The EUROSPAN and FHOCS cohorts did not contribute to the DIAGRAM data of the utilized publication[41].

*Pathway enrichment analysis.* We used GSA-SNP2 software for gene set enrichment analysis based on GWAS p-values[40]. This tool employs the Z-statistic of the random set model. We used a 20 kilobase window upstream and downstream of the gene for the SNP to gene annotation and removed adjacent genes highly correlated in the European population. We used pathway annotation from the MSigDB C2.CP (curated canonical pathways) version 5.2 database[100], therein the C2 canonical pathway database, which consists of 1329 curated gene sets that represent a biological process compiled by domain experts[101,102]. From this knowledge source, we selected pathways that were linked to T2D (previously published set of gold standard pathways for T2D[40]) and CVD (defined by us as pathways that were statistically significantly enriched in the CARDIoGRAM GWAS data (42,335 CVD cases and 78,240 controls))[103]. Pathways with a q-value < 0.25 were considered significantly enriched.

*Mendelian randomization.* We conducted a univariable two-sample MR study with Cer22:0 as phenotype and T2D as the outcome[41]. We only conducted an MR on the putative Cer22:0 effect on T2D risk because it was the only ceramide for which genome-wide suggestively significant SNPs were detected in EPIC-Potsdam and replicated in an independent study. We selected the SNP with the strongest Cer22:0 association in EPIC-Potsdam that was available in the replication datasets as instrumental variable for a univariable MR and harmonized the data for the direction of the effects between phenotype and endpoint associations. We used the R-packages 'TwoSampleMR' (v0.5.5) from the MR-Base platform[104] and "MendelianRandomization" (v0.5.0) to generate SNP specific Wald ratios (SNP-endpoint estimate divided by SNP-phenotype estimate) for the phenotype-endpoint associations.

*Mediation analyses.* We used the potential influence of red meat consumption and coffee consumption on T2D risk to explore the role of (dh)ceramides as potential mediators of lifestyle effects on cardiometabolic risk. These exposures were chosen

because they contributed to T2D-prediction beyond other established risk factors in the EPIC-Potsdam study[105–107], and the hypothesis that these exposures act through modification of lipid metabolism is biologically plausible. In a first step, we selected potential ceramide-mediators by regressing the food of interest on all T2D-related ceramides, adjusting for potential confounders [age, sex, T2D-related dietary exposure other than the exposure (from the set of red and processed meat, coffee, and whole grain), fasting status, total energy intake, leisure-time physical activity, medication (antihypertensive and lipid-lowering drugs), smoking (four categories, never, former, current < 20 Units per day, current > 20 Units per day), alcohol consumption, and education]. T2D-related ceramides were selected as potential mediators if they were statistically significantly and directionally consistently (one-sided p-value < 0.05) associated with the exposure.

Then, we estimated the proportion explainable (PE) as percentage attenuation of the association between exposure (food group or anthropometric trait) and outcome (T2D risk) in Cox models with adjustment for the selected ceramides compared to the same model without ceramide adjustment, using the delta method[24,108,109]. Bias corrected 95% bootstrap confidence intervals for the PE were constructed with the bcajack-function from the bcaboot package (CRAN.R-project.org/package=bcaboot) with 1000 replications and a two-thirds sampling fraction.

**Software.** Statistical Analysis System (SAS) Enterprise Guide 7.1 with SAS version 9.4 (SAS Institute Inc., Cary, NC, USA) was used to manage and prepare datasets and transform the lipid values. The pcalg-package in R (version 3.5.2 (20 December 2018)) and a developmental version of the NetCoupler-package (available from M.B.S. and C.W. upon request) were used to generate the metabolite networks and link them to disease incidences. QCtool v1.4 and SNPtest v2.5.2[110] were used for the GWAS on lipids. MR studies were conducted using the 'TwoSampleMR' (v0.5.5) from the MR-Base platform[104] and the "MendelianRandomization" (v0.5.0)[111] R packages in R (version 3.6.3 (29 February 2020)). Mediation analyses were conducted in R (version 3.5.2 (December 2018)).

**Reporting summary.** Further information on research design is available in the Nature Research Reporting Summary linked to this article.

## Data availability

The data that support the findings of this study are not publicly available due to data protection regulations. In accordance with German Federal and State data protection regulations, epidemiological data analyses of EPIC-Potsdam may be initiated upon an informal inquiry addressed to the PI of the EPIC-Potsdam study, who is the corresponding author of this manuscript [MBS]. Requests for data access are discussed in monthly EPIC-Potsdam investigator meetings, where data access proposals are either directly approved or adjustments of the proposal are requested to ascertain scientific soundness of EPIC-Potsdam data analysis. GWAS data from the EUROSPAN consortium are not deposited into publicly available databases in order to comply with individual cohort informed consent and participant data privacy restrictions. Specific data access requests can be granted through agreement by the cohort PIs following request, which can be addressed initially to Andrew Hicks, and which will usually be processed within 15 days. The utilized data from the FHSOC study were previously published[39]. A list with all analyzed ceramides and dihydroceramides along with LIPID MAPS-ID and HMDB-ID (if available) is provided as Supplementary Data 9. Metabolite sets for enrichment analysis were obtained from the MSigDB C2.CP database together with GSA-SNP2 tool (https://sourceforge.net/projects/gsasnp2/files/data/popular_pathway_data-20170227T151601Z-001.zip).

## Code availability

A generalized version (R-package) of the *NetCoupler-algorithm* can be accessed on GitHub (https://github.com/NetCoupler/NetCoupler). The development versions and settings used to generate the results herein are not on GitHub and are available on request from the corresponding author [M.B.S.].

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

## Acknowledgements

We thank the Human Study Centre (HSC) of the German Institute of Human Nutrition Potsdam-Rehbrücke for the processing of the biological samples and data management. We also thank the three consortia that contributed summary-level data for genetic associations with cardiometabolic endpoints: CARDIoGRAMplusC4D (http://www.cardiogramplusc4d.org/), MEGASTROKE (http://www.megastroke.org/), and DIAGRAM 471 (http://diagram-consortium.org/). The MEGASTROKE project received funding from sources specified at the website (http://www.megastroke.org/acknowledgments.html). This work was supported by a grant from the German Federal Ministry of Education and Research and the State of Brandenburg to the German Center for Diabetes Research (DZD grant 82DZD00302) and by a grant from the European Commission and the German Federal Ministry of Education and Research within the Joint Programming Initiative A Healthy Diet for a Healthy Life, as part of the ERA-HDHL cofounded joint call Biomarkers for Nutrition and Health (01EA1704). Clemens Wittenbecher was supported by the German Research Foundation's (DFG) individual fellowship #WI5132/1-1. Luke Johnston was funded through a Danish Diabetes Academy Postdoctoral Fellowship.

## Author contributions

C.W. led the conceptualization, formal analysis, methodology, software development, visualization of results, writing, reviewing, and editing of the manuscript, and contributed to the data curation. R.C. led the formal genome-wide association analyses, supported the conceptualization, data curation, methodology, software development, visualization of results, and reviewed and edited the manuscript. L.J. supported the methodology, contributed to the software development, and reviewed and edited the manuscript. F.E. supported data curation, formal analysis, methodology, contributed to the visualization of results, and reviewed and edited the manuscript. S.J. led the

Mendelian randomization analyses, supported the formal analysis, methodology, and reviewed and edited the manuscript. O.K. supported data curation, methodology, and reviewed and edited the manuscript. M.P. reviewed and edited the manuscript. F.D.G.M. and A.A.H. conducted the analyses of the EUROSPAN GWAS data and reviewed and edited the manuscript. P.H. contributed to generating the genetic data in EPIC-Potsdam and reviewed and edited the manuscript. J.K. supported conceptualization, methodology, software development and reviewed and edited the manuscript. F.B.H. contributed to the data interpretation and reviewed and edited the manuscript. M.B.S. led the funding acquisition, supervised the conceptualization and methodology, and reviewed and edited the manuscript.

## Funding

## Competing interests
The authors declare no competing interests.
