## [Peer Review File · Nature Communications]

Dihydroceramide- and ceramide-profiling provides insights into human cardiometabolic disease etiologyREVIEWER COMMENTS

Reviewer #1 (Remarks to the Author):

The manuscript by Wittenbecher et al. describes a comprehensive set of analyses of ceramide and dihydroceramide species in relation to cardiometabolic risk. This analysis is performed on a large case/cohort subset of the EOIC-Potsdam study (1886 samples) with 775 incident T2D and a second case/cohort subset (1707 samples with 551 incident CVD)

Lipidomics was performed by Metabolon measuring ceramide and dihydroceramide species, 27 in total. No information is provided on the assay performance in these analyses or indeed on how this was assessed. This is necessary to evaluate the data quality used in the subsequent analyses. What were the CVs across the analyses? How was this assessed?

Baseline data is provided for the cohort, but it appears not for the incidence cases of T2D and CVD. This should be provided and compared to the controls. HDL-C a biomarker for both T2D and CVD should also be included in this and as a covariate in subsequent analyses. This is particularly important as the adjustment for total lipid load is required in these analyses (as the authors have partially done using total cholesterol and triglycerides) however, HDL-C should be part of this.

As an aside, adjustment for lipid lowering drugs does not work in this context and may actually weaken the analyses. Statin lower total lipid load (to varying degrees depending on the lipid species, but typically not to normal/control levels. Adjustment for this treatment then tries to align with the control group and so effectively lowers the levels further. Whereas the adjustment should be factoring in the pretreatment levels which were higher. In my experience there is no easy way to adjust for this in lipidomic analyses and so exclusion of these or even no adjustment is probably better than including statin treatment as a covariate.

Supp Table 3 shows HR for T2D and CVD in minimally adjusted and fully adjusted models (no p-values are provided and no correction for multiple comparisons is performed. Whilst the significance (or lack thereof) after correction, does not preclude further analyses of these lipid associations, it should still be provided here.

As it is I am surprised that the associations appear so weak given the large numbers of control and cases in these analyses. Thus my concern about the lipidomic data quality. The authors then perform network adjusted modeling to identify the key species associated with T2D and CVD.

With the resulting subsets of lipid species, they perform GWAS and identify three gene regions ($p < 1E-7$), given the high p-value used here I think validation will be required. It is difficult to read Figure 2 but I can only see one region SPTLC3 that has associations $< 1E-7$, the others are higher p-values. This should be explained and p-values supplied in Table 2.

I think the enrichment of ceramide associated SNPs is difficult to justify and without any validation is likely to be presenting many false links to biology. It is not clear how many SNPs were used in this analyses and the use of a q-value cutoff of 0.25 (by my calculation) seem overly optimistic. I think these choices would require clear justification.

The subsequent Mendelian randomisation is also quite questionable with the relaxed p-value and having this based on a single SNP.

The mediation models do not provide p-values and again I suspect are now powered to detect these effects.

Overall I like the concept of this study and indeed am somewhat surprised that the results are not more convincing. However I have serious reservations with the analyses as presented given the limited power of the dataset.

Reviewer #2 (Remarks to the Author):

this is interesting work from an excellent group and certainly, the need for more understanding of pathways linking lifestyle factors etc and diet to diabetes and CVD would be useful - however, some important issues remain:

- what does posteriori mean - post hoc would be better - and does ref 14 support a significant interaction based on baseline ceramide levels that is independent of baseline higher risk in those with highest levels?

- line 72 - when you say "linked" what do you mean - correlation in observational studies? Good to be more specific for reader

ref 33 seems important but it is not complete?

- did you adjust for multiple comparisons given so many ceramides and by chance one or two could be considered to be significant? If not, why not?

- did you conduct pleiotropy checks in your MR analyses -

- is there any support for coffee intake or red meat from randomized trials and risk of diabetes or intermediate markers - this reviewer is not familiar with such and wonders if this could be confounded by other lifestyle factors if most of the evidence comes, as anticipated, from observational studies - note, in many diabetes studies, deprivation or measures of social class are not included and it is well known that lower SES eat more red meat and drink less coffee for example so this could be an upstream determinant of both disease risks and ceramide levels?

- there is not much evidence for influence of inflammation pathways on diabetes risk in big diabetes GWAS or do I have this wrong?

Overall, you present nice ideas but I think your work needs independent corroboration and more careful statistical analyses to adjusted for

1. multiple comparisons
2. other measures of pleiotropy in MR analyses
3. Social class in observational analyses

if you can do these, and validate findings in another cohort, then readers would be more willing to accept results but presently, there is quite a bit of speculation in this work as I read it - you have nicely and fairly acknowledged some of these limitations in your discussion but seems these are sticking points to a more robust analysis of your hypotheses.

Reviewer #3 (Remarks to the Author):

Summary: Overall, I found this to be an interesting and well-presented manuscript. The authors propose an important role for ceramides in cardiometabolic health (specifically type 2 diabetes and cardiovascular disease) and in linking diet to cardiometabolic health. Although there is existing literature in this area, the novelty that I believe is being claimed here relates to the number of (dh)ceramides being quantified and the consideration of the joint effect of these on the disease outcomes, i.e. the effect of each (dh)ceramide, conditional on all others (via implementation of the author's own NetCoupler algorithm, implemented in R). Furthermore, this paper provides evidence from a number of different though related analyses. More methodological detail is needed in places both to justify approaches taken and to ensure others could replicate the work. Although a little on the long side, the discussion provides a useful summary of the results in the context of existing literature and includes a useful consideration of potential limitations across the various methodologies used.

Major points

1) Methods – comparing the z-scores to the z-scores of log-transformed ceramide levels, I am wondering why the log transformation was necessary? Most of the distributions look pretty normal before the transformation. Relating to this, there is a statement in the Methods (p.25, L871-2) that

says “The (dh)ceramide concentrations were log-transformed to an approximately normal distribution and z-scaled so that all regression estimates are per 1 SD.” But this is a 1 SD change on the log-transformed scale, no? There needs to be clarity throughout (including table & figure legends) regarding the units/scale of the HR.

2) Methods (mediation) – I was wondering why you chose to estimate PE rather than estimate the indirect effect itself (e.g. using the difference in coefficients or product of coefficients methods)? Is there something about the model required that precludes these potentially more contemporary approaches? Also, I couldn’t find reference to the “delta method” in the references supplied.

3) Methods & Results (MR) – more details are needed here (in the interests of reproducibility and transparency). I would advise consulting the proposed STROBE-MR guidelines (available at: <https://peerj.com/preprints/27857v1/>). For example, details of the R package used to conduct the MR accounting for LD (including the version and specific functions used) are needed. Also, I can’t see the heterogeneity test results anywhere – although also not sure how valid this is with two SNPs? And did you consider using an independent (more powerful) GWAS to identify the ceramide instruments and/or for extracting betas? (See point (7) below)

4) Instinctively, the sample size (specifically the number of cases) in these analyses seems low. This is particularly true when you consider the number of covariates fitted in, for example, the Cox regressions whose results are presented in Supp. Table 3. There are 19 covariates in the HR2 model and (as I understand it) only 30 CVD cases (and 70 T2D cases). I was wondering what, if any, diagnostic checks you ran to check the reliability of the Cox regression model(s) in this context? Similarly, what could be the impact of this on the NetCoupler (and mediation) parts of the analysis? Is the only consequence limited power, or is there potential for spurious results? For example, in correlation analyses with low N, you can get high values of r but these are not generally reliable.

5) To what extent do you think you would have seen the same gene and/or pathway enrichment patterns (following the GWAS) for the subset of ceramides not associated with either of your disease outcomes? As currently presented, it is not clear to me that there is evidence that the subset of associated ceramides behave differently to those not associated.

6) Results, p.11 (MR results) – it would be useful to have the equivalent observational result shown in Table 3. Is there a way to get the observational Cox regression result and the MR result on a comparable scale?

7) In the limitations paragraph (p.15), you acknowledge that your GWAS had “limited statistical power”. Did you consider using GWAS results from elsewhere, for example, to identify instruments for the MR? As well as the two GWAS you refer to (Shin et al (2014) and Cresci et al (2020)) there are others that include at least some of your (dh)ceramides and have sample sizes ranging from 4,000 to 10,000, e.g. PMID 25378659 (Lemaitre et al 2015), PMID 22359512 (Demirkan et al 2012), PMID 19798445 (Hicks et al 2009).

8) In the limitations paragraph (p. 15), you say “pleiotropic effects through other ceramides cannot be excluded”. I would say there is a fair amount of evidence that such effects do in fact exist – both in your GWAS and in the papers mentioned above (point (7)). rs680379, for example, is associated with a number of different ceramides. In this context, it seems naïve to think you can use MR to isolate the causal effect of C22:0 from all other ceramides. In the early part of the paper you demonstrate the existence of a network linking various (dh)ceramides – it is hard to imagine these correlations/dependencies are not reflected in the genetics of these traits (ceramide levels). For this reason, I would suggest avoiding (or further contextualising) statements that single out Cer22:0 as having some kind of special/unique role to play in disease, such as “... Mendelian randomization studies supported a causal link of Cer22:0 with higher T2D risk ...” (P.13, L295).

9) To what extent do you think each of your analyses are robust to reverse causation, i.e. an effect of disease (sub-clinical in the case of the prospective association analyses) on ceramide levels?

Minor points

10) Fasting effect – it looks from table 2b that there is an effect of fasting, at least on the total dh-ceramide levels, if not the total ceramide levels. I understand you fit fasting status as a covariate in your models, so its effect on individual analyses is not in question. However, I am wondering to what extent you think the patterns or association/correlation you see in these mostly fasted (>8 hours) samples are indicative/representative of everyday function, i.e. non-fasted levels? For example, you talk in the discussion about ceramides concentrations acting as ‘nutrient sensors’ and being sensitive

to 'metabolic challenge'.

11) Abstract - It would be useful to have an indication of the sample size (and case/control balance) at least for the main observational work in the Abstract

12) Intro, p.2, L76 – “ceramides are among the most likely mediators of the effect of diet on cardiometabolic risk.” – this sounds like there is only one set of mediators and that they are ceramides. More likely is that there are many mediators in this relationship.

13) Intro, p.2, L91-94 – This final sentence in the intro. is hard to read/comprehend.

14) Methods – there appears to be no adjustment for multiple testing (i.e. for testing 25 (dh)ceramides in Suppl. Table 3). Please comment on this.

15) Methods, P.23, L786 – you mention a “standardized procedure” – could you add further details about the time between sample collection, processing and freezing? And also, the temperature the sample was kept at in between these steps? Perhaps this has been published elsewhere?

16) Methods, P.23-4 (Genetics) – did you check in the imputed data for any residual structure relating to the original genotyping batch (e.g. by running a GWAS for batch, and/or doing a PCA and looking for batch-related clustering)? If such structure existed and batch selections were based on, for example, case status for some disease, this could maybe create issues in the GWAS.

17) Methods, P24, L863-4 – it's not clear to me what the values in brackets represent. I thought it was the imputation value but the values don't really make sense (e.g. 12 for BMI). Perhaps it is the number of missing values that had to be imputed? Please clarify. Also, can you add more detail about the linear regression model used to determine the imputation values? What about when categorical data were missing?

18) Methods, P24, L864-5 – how was fasting status coded – is it a continuous trait (e.g. hours) or binary (categorisation based on more or less than 8 hours)?

19) Methods, P.25, L866 – what are the corresponding alcohol intake levels for the different categories, i.e. how are they defined?

20) Methods, P.26 (GWAS) – Plink v1.07 is listed in the software section but it's not clear where in the analysis it's used.

21) Methods, P.26 (GWAS) – please give details of the specific SNPTTEST model used. Also, there is no mention of the X chromosome – was data not available for this?

22) Methods, P.26 (GWAS) – is the GWAS also done on log-transformed z-scored data? Please clarify in the text.

23) Results – a useful addition might be a correlation heatmap to show the basic correlation structure. This is a figure that will be familiar to most people and may help to demonstrate the extra information captured by the proposed NetCoupler-algorithm.

24) Results – for ease of reading, please include sample and ceramide numbers throughout the results (not just in the methods). For example, in the first section, state the no. of (dh)ceramides quantified and the no. of participants (including by case/control status).

25) Results, P.6, L177-8 – as I understand it, the GWAS was only run for the ten (dh)ceramides listed in table 1 – please clarify this (if true) in the text.

26) Results, P.9, L230-2 – it's not clear to me what you mean by “GWAS signals below the genome-wide significant threshold”. Did you use the full results for this, or do you mean those with $P < 10^{-5}$ (i.e. your suggestive significance level)?

27) Results – Figure 3 – the quality of this figure needs improving as I can't make out the gene names on the Manhattan plots. Also, the legend sentence that begins “Shown are pathways ..” doesn't make sense to me.

28) Results (mediation) – the PE's seem high. Is this expected? To what extent could this be capturing effects of correlated ceramides.

29) It is not clear to me why the MR was only done for one of the SNP/ceramide pairs in Table 2. Is it because you wanted more than one instrument? Or because these SNPs weren't in the disease GWAS summary stats you were using? Further clarity is required on this point.

Typos, grammar, etc.

30) Intro, p.2, L56/57 – “and thereby base prevention efforts” – this sentence doesn't make complete sense.

31) Intro, p.2, L83 – “encoded in metabolomics network” – grammar issue here.

Reviewer #4 (Remarks to the Author):

This is an interesting paper that throws several different causal inference techniques at the question of whether ceramides causally effect risk of T2D and CVD. This is a difficult – perhaps impossible – question to definitively answer with observational data. The authors make a noble attempt, and their results, together with experimental data from animals, are somewhat convincing – although I'm not ready to start using ceramides as surrogate markers for T2D or to believe that intervening on ceramides will decrease risk of T2D. A few specific comments are below:

1. Comments about the NetCoupler-algorithm:

The NetCoupler-algorithm seems to be an empirical way of creating a network of ceramides based on their conditional correlations, and then investigating the direct effect of a specific ceramide on the outcome of interest (e.g., T2D), by seeing if its association with the outcome remains after fitting separate models adjusting for all possible combinations of the other adjacent variables. The thought is that if the ceramide has a direct effect on the outcome, then it will remain associated with the outcome whether one adjusts for other ceramides on the causal pathway and/or other ceramides that could be potentially confounding the relationship.

1a. The NetCoupler algorithm appears that it may be sensitive to the order in which the ceramides are considered (because those found to have direct effects are included as potential confounders in subsequent models). How robust are the selected ceramides to the order in which they were considered?

1b. Were Cox models properly weighted to account for the over-sampling of cases? This is not clear.

1c. Direct effects seem to be of interest, but total effects might be of more interest. If one ceramide leads to a cascading effect of other ceramides that then leads to T2D, you would want to capture the total effect of that ceramide. If the total effect is only through the cascade (indirect effect), then one would miss this ceramide, even though it is causing T2D.

1d. Removing Cer26:1 because it was non-significant in the joint model is somewhat unappealing and sort of runs counter to the authors' rationale for using the NetCoupler algorithm in the first place. Its non-significance must imply that one (or some) of the other ceramides are confounders or mediators between its relationship with T2D. But that was ruled out in the original NetCoupler selection process, right? Non-significance is a crude measure of the importance of a variable.

2. There seems to be a lot of reliance throughout the manuscript on significant p-values.

3. Comments about Mendelian randomization:

3a. Based on Mendelian randomization, Cer22:0 is said to be causally associated with T2D (estimate = 0.068, $p=0.031$) but not with CAD (estimate = -0.052, $p=0.20$). P-values are dependent on the number of outcomes. There were more T2D outcomes (775) than CAD outcomes (283, I think). Perhaps with more outcomes / bigger sample size, Cer22:0 would have crossed the magical 0.05 p-value threshold. (Again, this is a problem with reliance on statistical significance to infer causality.) However, a counter-argument to what I just said was that the CAD and stroke estimates went in opposite directions, which might be difficult to explain if there truly is a causal effect.

3b. For what it's worth, I am skeptical of almost all Mendelian randomization studies. I have a hard time believing that all of the assumptions are met – particularly that the direct effect of the SNP on the outcome is only through its effect on the ceramide. VanderWeele et al., (2014) have a nice paper on some of the challenges of using this approach to infer causality (<https://pubmed.ncbi.nlm.nih.gov/24681576/>).

4. Comments on Discussion:

4a. Consistency with animal results is reassuring.

4b. Use as surrogate endpoints is a little premature/ over-interpreting results, especially given the relatively small HR for most of the ceramides.

4c. I appreciated the authors discussion of their limitations – most of the limitations are things that I had identified while going through the manuscript (and are written perhaps a little more strongly above), and I agree with the limitations.

5. Style: I'm not a regular reader of Nature Communication, so perhaps some of these critiques are due to the way that papers for this journal are often formatted, but I found myself flipping back and forth frequently between Results and Methods. My understanding is that essential information is supposed to be included in Results and more technical information is included in Methods. However, I felt like essential information such as sample sizes of the cohorts, the timing of the measurement of ceramide concentrations, number of T2D and CVD events should have been included in the Results.

6. Other comments:

6a. Line 749: Censoring date of 31st of November, 2006 does not exist.

6b. The density plots of the z-scores and the z-scores of the log-transformed variables (Figure 1A), are not very helpful for determining whether the normality assumption now holds. A better visualization is needed.

6c. "Comprehensive confounder adjustment" is a misnomer. You can never know if you have comprehensively adjusted for all confounders.

We thank all reviewers for the exceptionally thoughtful and valuable comments. We have substantially revised the manuscript. To facilitate readability, all the references to pages (p) and lines (L) in the response letter refer to the clean version of the manuscript (without tracked changes).

Reviewer #1 (Remarks to the Author):

The manuscript by Wittenbecher et al. describes a comprehensive set of analyses of ceramide and dihydroceramide species in relation to cardiometabolic risk. This analysis is performed on a large case/cohort subset of the EOIC-Potsdam study (1886 samples) with 775 incident T2D and a second case/cohort subset (1707 samples with 551 incident CVD)ⁱ

Lipidomics was performed by Metabolon measuring ceramide and dihydroceramide species, 27 in total. No information is provided on the assay performance in these analyses or indeed on how this was assessed. This is necessary to evaluate the data quality used in the subsequent analyses. What were the CVs across the analyses? How was this assessed?

Response: *We did a pilot study in 35 EPIC-Potsdam participants assessing the intra-class correlation coefficients (ICC) of the included ceramides and dihydroceramides. Lipidomics profiling was conducted in the blood samples from both time points, and the ICCs for all (dh)ceramides were computed, which relate the between-person variance to the total variance (between-person variance + within-person variance). The ICC is an integrated measure of technical and biological variance. To this end, blood samples were taken approximately six weeks apart. Ceramides C14:0, C18:1, and C26:1 had ICCs around 0.4, suggesting poor to fair reliability; however, most ceramides had ICCs above 0.5 (mostly, above 0.6, good, or above 0.75, excellent reliability). Compared with the ceramides, the dihydroceramide measurements were less reliable. Six dh-ceramides (C16:0, C20:0, C22:1, C24:0, and C24:1) had fair ICCs, while ICCs for the remaining six dh-ceramides suggested substantial biological variance and, consequently, poor reliability. These results indicated that most (dh)ceramides were applicable for epidemiological risk assessment. We now describe the findings from this pilot study in the results section ('In a pilot study in 35 EPIC-Potsdam participants with two blood samples taken approximately six weeks apart, we assessed the within-person agreement of (dh)ceramide-measurements. The intraclass correlation coefficients (ICC) from the pilot indicated fair to excellent reliability of most ceramide- and about half of the dihydroceramide-measurements. However, few ceramide- and about half of the dihydroceramide-measurements showed poor reliability (Supplementary Figure 1).') (p4, L100-105) and added Supplemental Figure 1.*

The ICCs of the detected disease-related (dh)ceramides indicated fair to excellent reliability. We found no disease-associated (dh)ceramides with poor reliability, which is expected because poor reliability introduces noise. Moreover, we randomly distributed the samples across batches. Therefore, the technical variance is expected to be unrelated to the case status. Completely random noise is expected to bias estimates towards the null. In the revised version, we discuss the potential limitations of the lipidomics measurements: 'We measured the (dh)ceramide plasma concentrations at a single time point. Some of the (dh)ceramides showed strong within-person variance in a pilot study with repeated measurements after several weeks. Under the assumption that the introduced variance is unrelated to the disease risk, poor reliability is expected to bias risk estimates towards the null. Accordingly, (dh)ceramides with low ICCs were not selected as disease risk markers, whereas the disease-associated (dh)ceramides had fair to excellent ICCs in our reliability study.' (p18, L428-434)

Baseline data is provided for the cohort, but it appears not for the incidence cases of T2D and CVD. This should be provided and compared to the controls.

***Response:** The case-cohort design is, in some regards, fundamentally different from the perhaps more common case-control study. The subcohort is a representative subsample of the full cohort (more specifically, a random subsample of at-risk-participants who are free of the endpoints at baseline). It includes participants with the incident disease randomly, resembling the incidence rate of the endpoint in the full cohort. Therefore, the distribution of baseline variables in the subcohort is representative for the distribution of these variables in the full cohort. The oversampling of participants with the incident disease increases the power to detect prospective associations. With appropriately weighted models, the case-cohort design generates unbiased relative risk estimates for the full cohort.*

*The recommended descriptive statistics for case-cohort studies is showing the distribution of covariables (i.e., potential confounders) over exposure categories in the subcohort, which is representative for the distribution of these variables in the full cohort. We follow these recommendations by displaying the covariables across ceramide distribution-based quintile categories (**Supplemental Table 2a**), and across dihydro-ceramide distribution-based quintile categories (**Supplemental Table 2b**). The study design does not include controls, but the random subcohort with internal cases, and external cases that are differently weighted in the prospective analyses. A "case-control"-comparison does not correspond to the study design.*

HDL-C a biomarker for both T2D and CVD should also be included in this and as a covariate in subsequent analyses. This is particularly important as the adjustment for total lipid load is required in these analyses (as the authors have partially done using total cholesterol and triglycerides) however, HDL-C should be part of this.

***Response:** We have added sensitivity analyses that show the additional adjustment for HDL does not have substantial influence on the risk estimates of the disease risk association of the selected (dh)ceramides (**Supplemental Table 6**). Besides adjustments for cholesterol and triglycerides in the main analysis, all final models were also adjusted for total ceramide and total dihydroceramide concentrations.*

As an aside, adjustment for lipid lowering drugs does not work in this context and may actually weaken the analyses. Statin lower total lipid load (to varying degrees depending on the lipid species, but typically not to normal/control levels. Adjustment for this treatment then tries to align with the control group and so effectively lowers the levels further. Whereas the adjustment should be factoring in the pretreatment levels which were higher. In my experience there is no easy way to adjust for this in lipidomic analyses and so exclusion of these or even no adjustment is probably better than including statin treatment as a covariate.

***Response:** Adjustment for use of lipid lowering medication as a binary variable in a multi-variable Cox model allows the baseline risk to vary between the two groups (with and without pharmacological treatment), assuming fixed effects of the exposure (same slope of the exposure-outcome relation) across the groups. As you explained, statins influence both, the concentrations of the individual lipids and the disease risk (as demonstrated in trials), corresponding to the definition of a potential confounder. Therefore, adjustment for lipid lowering medication should not bias the estimates but block confounding. We would suggest keeping lipid lowering medication as a covariable in the fully adjusted models. To assess whether lipid lowering medication may modify the (dh)ceramide-effects on disease risk, we have included sensitivity analysis excluding the participants on lipid-lowering medication at baseline (**Supplemental Table 7**).*

Concerning the two comments above, we state the revised version of the manuscript: 'In sensitivity analyses, neither additional adjustment of the final model for HDL-cholesterol (**Supplemental Table 6**) nor exclusion of participants on lipid-lowering medication at baseline (**Supplemental Table 7**) did substantially change the effect estimates for T2D risk or for CVD risk.' (p6, L177-179)

Supp Table 3 shows HR for T2D and CVD in minimally adjusted and fully adjusted models (no p-values are provided and no correction for multiple comparisons is performed. Whilst the significance (or lack thereof) after correction, does not preclude further analyses of these lipid associations, it should still be provided here.

Response: Done.

As it is I am surprised that the associations appear so weak given the large numbers of control and cases in these analyses. Thus my concern about the lipidomic data quality. The authors then perform network adjusted modeling to identify the key species associated with T2D and CVD.

Response: In general, an association between and ceramide levels and CVD risk is indeed established. However, our main models are not only extensively adjusted for potential confounders including environmental factors, anthropometry, and standard blood lipid markers but also for total ceramide- and dihydroceramide-levels. Particularly for CVD, this additional adjustment clearly attenuated the risk associations of several single (dh)ceramides, which explains some difference between our findings and published associations. To show the effect of specific adjustments more comprehensible, we included an additional model in the revised **Supplemental Table 3**, which is extensively adjusted for potential confounders but not for total ceramide and dihydroceramide levels. Several ceramides and dihydroceramides were statistically significantly associated with CVD risk in this model.

We must also admit that our minimal adjusted models that related single (dh)ceramide levels with CVD risk had a coding error. We included exact age not age in years as strata variable in the Cox models. After correcting this, more significant risk associations were detected. Importantly, this only applies to some of the single-ceramide models in Supplemental Table 3, not to our main analyses. Thank you for asking for the revision of this supplementary information!

Also, we do not fully agree that the ceramide endpoint association in the final models are throughout weak, or weaker than expected. Several (dh)ceramides were statistically significantly associated with cardiometabolic risk after very extensive confounder adjustment, including adjustment for total ceramide and dihydroceramide levels. For example, more than doubling of T2D-risk per SD higher CER22:0 level is a very marked effect. We randomized samples across batches, any technical variance is not expected to be related to the case status. Several strong associations were detected. Therefore, we do not think that our statistical analyses indicate general technical issues with the (dh)ceramide measurements.

With the resulting subsets of lipid species, they perform GWAS and identify three gene regions (p<1E-7), given the high p-value used here I think validation will be required. It is difficult to read Figure 2 but I can only see one region SPTLC3 that has associations <1E-7, the others are higher p-values. This should be explained and p-values supplied in Table 2.

Response: Thank you very much for this feedback. Based on this along with other comments, we have decided to include additional data from the EUROSPAN consortium, and to compare our results with published Gwas from the Framingham Offspring Cohort. We included these external GWAS data in the main Tables restructured the presentation of our GWAS results. As suggested, the revised **Table 2** shows

more details about the significance of single SNPs, including all the results that were available for validation with GWAS data from other cohorts. We emphasize the replicated SNP-(dh)ceramide associations as our main GWAS results. Accordingly, we moved the old Figure 2 to the Supplement (now **Supplementary Figure 3**). However, the coverage of (dh)ceramide in other studies was more limited than in our study, and, therefore, replicability could not be evaluated for the GWAS on most (dh)ceramides. In the revised version, we provide comprehensive information on the suggestive findings from all our GWAS in a searchable file to facilitate replication for future studies into genetic influence on plasma (dh)ceramide concentrations.

I think the enrichment of ceramide associated SNPs is difficult to justify and without any validation is likely to be presenting many false links to biology. It is not clear how many SNPs were used in this analyses and the use of a q-value cutoff of 0.25 (by my calculation) seem overly optimistic. I think these choices would require clear justification.

Response: *FDR=0.25 means that 75% of your hits are expected to be true positives. For a general screening of function, that is generally considered as a justifiable base to generate biological hypotheses, and a Q-value of 0.25 is frequently used in gene set enrichment analyses.*

Accordingly, the same cutoff (Q-value of 0.25) was suggested to select significantly enriched gene sets in the paper that introduced the method that we applied for our enrichment analyses (1). Yoon et al. showed in their paper, for example, that the use of this cutoff with data from T2D-GWAS led to unsupervised selection of gene sets that are known to be linked with T2D.

*However, we agree that the generalizability of our results is difficult to evaluate with GWAS data from a single study. The replication of the gene set enrichment results would have required full GWAS results on the same phenotypes generated with the exact same workflow. Such data were not available. Therefore, we moved the results from the gene set enrichment analyses to the supplement (**Supplementary Figure 4**) and only interpret some general patterns, clearly emphasizing the suggestive nature of these results.*

The subsequent Mendelian randomisation is also quite questionable with the relaxed p-value and having this based on a single SNP.

Response: *We restricted the MR to instruments in the SPTLC3 gene region that encodes a protein that is part of the rate-limiting enzyme in sphingolipid biosynthesis. We detected the strong association between SNPs in that region and Cer22:0 plasma levels in a genome-wide screen, and our observations were consistent with two other GWAS in independent study populations (2, 3). We now used the estimates for the SNP-CER22:0 association from these studies to replicate the MRs for the CER22:0 risk associations with T2D and CVD (**Table 3**).*

The mediation models do not provide p-values and again I suspect are now powered to detect these effects.

Response: *The mediation models generate confidence intervals based on bootstrapping, which is an established method for non-parametric estimation of confidence intervals of statistical estimates. Compared to parametric confidence intervals, the validity of bootstrap confidence depends on less assumption on the underlying distributions. As for many non-parametric estimations, p-values are not generated. However, a confidence interval for the proportion explainable that does not include the 0 translates into a type I-error probability below 5%. The lower confidence level of the estimated PEs is not even close to zero.*

Overall I like the concept of this study and indeed am somewhat surprised that the results are not more convincing. However I have serious reservations with the analyses as presented given the limited power of the dataset.

***Response:** Thank you very much for your valuable feedback. We hope that we alleviated your reservations in the revised version of our manuscript. We have addressed the raised issues with additional data on the measurement quality, additional sensitivity analyses, and external genetics data. The added results support the validity of our observational estimates, as well as the emphasized GWAS and MR results.*

However, for several (dh)ceramides GWAS data for replication were not available. Therefore, we moved some results to the supplement and toned down the interpretation. Nonetheless, these results will be a valuable resource for future studies into the genetic influences on plasma sphingolipid concentrations.

Reviewer #2 (Remarks to the Author):

this is interesting work from an excellent group and certainly, the need for more understanding of pathways linking lifestyle factors etc and diet to diabetes and CVD would be useful - however, some important issues remain:

- what does posteriori mean - post hoc would be better - and does ref 14 support a significant interaction based on baseline ceramide levels that is independent of baseline higher risk in those with highest levels?

Response: Indeed, post hoc is the correct term. The cited analysis of the PREDIMED trial as observational cohort showed that the level of some ceramides and a ceramide score were associated with higher CVD risk. Interaction analyses detected a significant interaction between the ceramides and intervention group on CVD risk. Ceramide levels were strongly associated with higher CVD risk in participants in the control group, whereas the risk association was attenuated with a Mediterranean diet intervention. We have rephrased the sentence: Besides, a post hoc analysis of the PREDIMED trial suggested that CVD prevention with a Mediterranean diet intervention particularly alleviated the higher risk of major cardiovascular events in participants with elevated ceramide levels before the intervention (14) (4)' (p3, 70-72)

- line 72 - when you say "linked" what do you mean - correlation in observational studies? Good to be more specific for reader

Response: We have changed the wording: 'Epidemiological studies have shown associations of ceramides and dihydroceramides with CVD and T2D risk' (p3, 64-65)

ref 33 seems important but it is not complete?

Response: Completed.

- did you adjust for multiple comparisons given so many ceramides and by chance one or two could be considered to be significant? If not, why not?

Response: Our analytical design deviates to some extent from most studies that test multiple exposure-endpoint associations. While we indeed tested several exposures, we run multiple models for each exposure, adjusting for all possible combinations of direct neighbors in the (dh)ceramide network [between 8 and 128 models per (dh)ceramide]. If in any of these models the p-value for the association of the exposure-ceramide with T2D risk was non-significant we did not reject H0. In other words, we did not classify the (dh)ceramide as having a direct on T2D risk if any of the multiple models generated a non-significant effect estimate. Finally, we tested whether all the selected (dh)ceramides with direct effect were significantly associated with the endpoint when simultaneously included into the same confounder-adjusted Cox-model. In this regard, the multiple testing problem is reversed: multiple tests of the same hypothesis increase the type II error probability. Reasoning that the robustness across multiple differentially adjusted models is expected to reduce the likelihood of chance-findings, we, therefore, did not further adjust for multiple testing.

- did you conduct pleiotropy checks in your MR analyses –

Response: Tests for directional pleiotropy, for example MR-Egger, depend on the use of several instruments that are not in high LD. However, we used single instruments from one genetic region. In the revised manuscript we show that use of the SNP-phenotype associations from different studies generated very consistent results. The MR instruments were in the SPTLC3 gene region. The encoded protein that is part of the rate-limiting enzyme in sphingolipid biosynthesis, providing a direct biological link between the instruments and the phenotype. Please, also see the answer to your comment on pleiotropy below.

- is there any support for coffee intake or red meat from randomized trials and risk of diabetes or intermediate markers - this reviewer is not familiar with such and wonders if this could be confounded by other lifestyle factors if most of the evidence comes, as anticipated, from observational studies - note, in many diabetes studies, deprivation or measures of social class are not included and it is well known that lower SES eat more red meat and drink less coffee for example so this could be an upstream determinant of both disease risks and ceramide levels?

Response: Coffee and red meat have both been associated with T2D risk in multiple observational studies that accounted for potential confounding by SES (5-11). The effect of red meat and coffee on type 2 diabetes in intervention trials is difficult to assess, depending on duration and control diet. However, several controlled dietary interventions show beneficial effects of red meat withdrawal and increase of coffee consumption on diabetes-related biomarkers (12-16). To generate functional biomarkers for diabetes-related metabolic pathways that may help to clarify if and how specific foods causally affect T2D risk was one of the aims of this study. We clearly acknowledge that our mediation analyses are of 'hypothesis-generating' (p3, L94) nature. However, testing the influence of selected foods on the proposed disease-related (dh)ceramides through dietary interventions is straightforward, while demonstrating the effect of single foods on hard endpoints in randomized trials is hardly possible.

- there is not much evidence for influence of inflammation pathways on diabetes risk in big diabetes GWAS or do I have this wrong?

Response: Low-grade inflammation is linked to T2D risk through several lines of evidence (17). The genetic underpinnings of this claim are summarized, for example, here (18).

Overall, you present nice ideas but I think your work needs independent corroboration and more careful statistical analyses to adjusted for

1. multiple comparisons

Response: As discussed in more detail above, we use the p-values from multiple models as evidence against rejecting H_0 . This requirement of robust effects across multiple differentially adjusted models should have reduced the probability of false positive associations.

2. other measures of pleiotropy in MR analyses

Response: We restricted the MR to instruments in the SPTLC3 gene region that encodes a subunit of the rate-limiting enzyme in sphingolipid biosynthesis. We detected the strong association between SNPs in that region and Cer22:0 plasma levels in a genome-wide screen, and our observations are consistent with two other GWAS in independent study population. In the revised version, we replicated the MR with the SNP-phenotype associations from the other cohorts, rendering consistent results. Therefore, the selected SNPs and MR estimates were consistently and biologically plausibly connected to the phenotype.

Moreover, we carefully evaluated mutually adjusted effects of a broad range of ceramides on the cardiometabolic endpoints. For Cer22:0 particularly, we showed that the association with high T2D risk was robust against adjustment for other (dh)ceramides. Additionally, none of the other T2D-related ceramides or dh-ceramides was similarly associated with SNPs in the SPTLC3 region. While not an off-shelf method, we do believe that our design provides arguments against attributing the described association structures to directional pleiotropy.

3. Social class in observational analyses

Response: We already used highest educational attainment in all our models to adjust for the influence of socioeconomic factors on cardiometabolic outcome. This is the variable that was associated with cardiometabolic risk and has been used as a proxy for the influence of socioeconomic factors in many previous EPIC-Potsdam publications.

Conceptually, along with other lifestyle factors, the composition of the diet is probably one of the downstream mediators of the influence of SES on cardiometabolic endpoints.

if you can do these, and validate findings in another cohort, then readers would be more willing to accept results but presently, there is quite a bit of speculation in this work as I read it - you have nicely and fairly acknowledged some of these limitations in your discussion but seems these are sticking points to a more robust analysis of your hypotheses.

Response: Thank you very much for your valuable feedback. We have addressed many of the raised issues with additional data on the measurement quality, additional sensitivity analyses, and external genetics data. The added results to support the interpretation of our observational estimates, as well as the emphasized GWAS and MR results.

However, we did not find data to replicate the GWAS on several of the (dh)ceramides, and the gene set enrichment analysis. Therefore, we moved these results to the supplement and toned down the interpretation. We think that these results will be a valuable resource for future studies on the genetic influences on plasma sphingolipid concentrations.

Reviewer #3 (Remarks to the Author):

Summary: Overall, I found this to be an interesting and well-presented manuscript. The authors propose an important role for ceramides in cardiometabolic health (specifically type 2 diabetes and cardiovascular disease) and in linking diet to cardiometabolic health. Although there is existing literature in this area, the novelty that I believe is being claimed here relates to the number of (dh)ceramides being quantified and the consideration of the joint effect of these on the disease outcomes, i.e. the effect of each (dh)ceramide, conditional on all others (via implementation of the author's own NetCoupler algorithm, implemented in R). Furthermore, this paper provides evidence from a number of different though related analyses. More methodological detail is needed in places both to justify approaches taken and to ensure others could replicate the work. Although a little on the long side, the discussion provides a useful summary of the results in the context of existing literature and includes a useful consideration of potential limitations across the various methodologies used.

Response: Thank you for your interest and the nice summary of our work.

Major points

1) Methods – comparing the z-scores to the z-scores of log-transformed ceramide levels, I am wondering why the log transformation was necessary? Most of the distributions look pretty normal before the transformation. Relating to this, there is a statement in the Methods (p.25, L871-2) that says "The (dh)ceramide concentrations were log-transformed to an approximately normal distribution and z-scaled so that all regression estimates are per 1 SD." But this is a 1 SD change on the log-transformed scale, no? There needs to be clarity throughout (including table & figure legends) regarding the units/scale of the HR.

Response: We aimed to report effect sizes that were comparable between the different (dh)ceramides. If established clinical cut points are still missing the per-SD-effects are an accepted way to convey comparable effect sizes relative to the population distribution of a set of biomarkers. Furthermore, if population distributions are skewed it is indicated to transform the biomarker distributions and derive the SDs on the a scale on which the distributions are approximately normal. For metabolite data we have previously shown that they are log-normal [see "Statistical analyses" section and reference to Suppl Material 10 in Krumsiek et al. (19)].

We derived the standard deviation (SD) on the log-scale because, like many circulating molecules, the non-transformed ceramide- and dihydroceramide concentrations tended to be right-tailed. Therefore, the log-transformation was indicated to avoid biased estimation of the SDs. We have revised Figure 1 to make the distributions easier to examine, and we have revised the description in the Methods section ('The (dh)ceramide concentrations tended to be right-tailed. Therefore, we log-transformed (dh)ceramide concentrations, which resulted in approximately normal distributions, and z-scaled the log-transformed values. Accordingly, all regression estimates were reported per 1 SD.'). **(p23, L647-649)**

However, the log-transformation is a monotone transformation, not affecting the order of the data. Therefore, 1 SD increment on the log-scale can be interpreted relative to a standard normal distribution. For example, a contrast of 1 SD on the log-scale corresponds to a comparison of a person with the median concentrations to one at the 84th percentile of the distribution; or 84th percentile vs. 97th; or 16th percentile vs. median, etc. However, this interpretation of per-SD-effects in terms of the corresponding

population distribution percentile comparisons is only valid if the SD was derived on the log-scale (on which our molecular data happened to be approximately normally distributed).

2) Methods (mediation) – I was wondering why you chose to estimate PE rather than estimate the indirect effect itself (e.g. using the difference in coefficients or product of coefficients methods)? Is there something about the model required that precludes these potentially more contemporary approaches? Also, I couldn't find reference to the "delta method" in the references supplied.

Response: Tyler VanderWeele (2011) showed that the difference in HR between a mediator-adjusted vs. non-mediator-adjusted Cox model on a log scale can be used to estimate natural indirect effects when the outcome is rare (20). This is what we do. We would still suggest keeping the "Proportion Explainable" as label because the definition is not as narrow as for indirect effect. Our analyses support the hypothesis that the diet-related FA-composition in ceramides could reflect biological mechanisms that may help to explain how specific foods influence cardiometabolic risk. Our results also suggest importance of some specific ceramides e.g., Cer18:0 for red meat effects on T2D, and dhCer22:2 for coffee effects on T2D. This might be useful for the design of future studies that aim to "explain" how these foods are related to T2D risk.

The strict counterfactual interpretation of an indirect effect (the difference of disease risk in the exposed when holding ceramide levels constant at the levels of the non-exposed) is experimentally not accessible and depends on numerous non-testable assumptions.

3) Methods & Results (MR) – more details are needed here (in the interests of reproducibility and transparency). I would advise consulting the proposed STROBE-MR guidelines (available at: <https://peerj.com/preprints/27857v1/>). For example, details of the R package used to conduct the MR accounting for LD (including the version and specific functions used) are needed. Also, I can't see the heterogeneity test results anywhere – although also not sure how valid this is with two SNPs? And did you consider using an independent (more powerful) GWAS to identify the ceramide instruments and/or for extracting betas? (See point (7) below)

Response: According to the comments, we have revised the MR analyses. The description of the applied methods now reads as follows:

'We conducted a univariable two-sample MR study with Cer22:0 as phenotype and the outcomes T2DM (21), CAD (22), and stroke (23). We decided to only conduct MR on the Cer22:0-endpoint associations because it was the only ceramide for which genome-wide significant SNPs were detected in EPIC-Potsdam and replicated in at least one independent study. We selected the SNP with the strongest Cer22:0 association as instrumental variable for a univariable MR, and used the R-packages 'TwoSampleMR' (v0.5.5) from the MR-Base platform (24) and 'MendelianRandomization' (v0.5.0) to generate IV-estimates for the phenotype-endpoint association.' (p24, L719-730)

As you suspected, tests for directional pleiotropy, for example, MR-Egger, depend on the use of several instruments that are not in high LD. However, we used single instruments from one genetic region. In the revised manuscript we show that use of the SNP-phenotype associations from different studies generated very consistent results. The MR instruments were in the SPTLC3 gene region. The encoded protein that is part of the rate-limiting enzyme in sphingolipid biosynthesis, providing a direct biological link between the instruments and the phenotype. Our observational analyses show that the association of Cer22:0 with

high T2D risk was robust against adjustment for other (dh)ceramides. Additionally, none of the other T2D-related ceramides or dh-ceramides was similarly associated with SNPs in the SPTLC3 region. While not an off-shelf method, we believe that our design provides arguments against attributing the described association structures to directional pleiotropy.

4) Instinctively, the sample size (specifically the number of cases) in these analyses seems low. This is particularly true when you consider the number of covariates fitted in, for example, the Cox regressions whose results are presented in Supp. Table 3. There are 19 covariates in the HR2 model and (as I understand it) only 30 CVD cases (and 70 T2D cases). I was wondering what, if any, diagnostic checks you ran to check the reliability of the Cox regression model(s) in this context? Similarly, what could be the impact of this on the NetCoupler (and mediation) parts of the analysis? Is the only consequence limited power, or is there potential for spurious results? For example, in correlation analyses with low N, you can get high values of r but these are not generally reliable.

Response: We applied the case cohort design. The number of cases is not 70 for T2D and 30 for CVD. These are the cases that were randomly included in the random subsample of the full cohort (the subcohort), the so-called internal cases. However, the case cohort also includes all other cases that occurred in the full cohort until censoring date. By statistically accounting for the oversampling of cases, this design derives unbiased relative risk estimate for the full cohort. In total, our study included 775 incident T2D cases and 551 incident CVD cases.

5) To what extent do you think you would have seen the same gene and/or pathway enrichment patterns (following the GWAS) for the subset of ceramides not associated with either of your disease outcomes? As currently presented, it is not clear to me that there is evidence that the subset of associated ceramides behave differently to those not associated.

Response: Among a set of intercorrelated molecular markers, the advantage of the NetCoupler-analyses, from our perspective, is precisely to differentiate between direct effects and correlated bystanders. For these intercorrelated phenotypes, some overlap between the GWAS results and the downstream analyses (MR, gene set enrichment) must be expected. However, the evidence from animal models indicates that the potential adverse metabolic role of (dh)ceramides depends on the contained acyl-chain. The network-adjusted observational analyses suggest that potential genetic influences are rather mediated by the disease-associated (dh)ceramides. We think that a potential overlap of the genetic influences on disease-associated (dh)ceramides and on non-disease-associated (dh)ceramides does not call into question our results or their interpretation herein.

6) Results, p.11 (MR results) – it would be useful to have the equivalent observational result shown in Table 3. Is there a way to get the observational Cox regression result and the MR result on a comparable scale?

Response: This request conflicts to some extent with comments from the other reviewers, rather suggesting to focusing less on the results from the MR. For example, reviewer 4 asks in comment 3b to consider a paper by Tyler VanderWeele about methodological challenges in MR studies, where specifically issues with the derivation of quantitative effect estimates are raised (25). We have considered these points by avoiding strong causal terminology in relation to the MR-estimates, and we would suggest to rather interpret the MR results as another hint to a biological link between specific ceramides and T2D risk, while refraining from interpretation of the quantitative MR estimates.

7) In the limitations paragraph (p.15), you acknowledge that your GWAS had "limited statistical power". Did you consider using GWAS results from elsewhere, for example, to identify instruments for the MR? As well as the two GWAS you refer to (Shin et al (2014) and Cresci et al (2020)) there are others that include at least some of your (dh)ceramides and have sample sizes ranging from 4,000 to 10,000, e.g. PMID 25378659 (Lemaitre et al 2015), PMID 22359512 (Demirkan et al 2012), PMID 19798445 (Hicks et al 2009).

Response: Thank you very much for this advice. We have included data from some of the suggested studies, including a look-up study in non-published data from Andrew Hicks' study, who is now a co-author on our paper. However, the coverage of (dh)ceramide in these studies was more limited than in our study, and, therefore, replicability could not be evaluated for the GWAS on most (dh)ceramides. In the revised version, we are still more careful with the interpretation of these suggestive findings. Additionally, we now provide comprehensive information on the suggestive findings from all our GWAS to facilitate replication for future studies into genetic influence on plasma (dh)ceramide concentrations.

8) In the limitations paragraph (p. 15), you say "pleiotropic effects through other ceramides cannot be excluded". I would say there is a fair amount of evidence that such effects do in fact exist – both in your GWAS and in the papers mentioned above (point (7)). rs680379, for example, is associated with a number of different ceramides. In this context, it seems naïve to think you can use MR to isolate the causal effect of C22:0 from all other ceramides. In the early part of the paper you demonstrate the existence of a network linking various (dh)ceramides – it is hard to imagine these correlations/dependencies are not reflected in the genetics of these traits (ceramide levels). For this reason, I would suggest avoiding (or further contextualising) statements that single out Cer22:0 as having some kind of special/unique role to play in disease, such as "... Mendelian randomization studies supported a causal link of Cer22:0 with higher T2D risk ..." (P.13, L295).

Response: In line with our answers to your comment above and other comments about potential directional pleiotropy in the MR analyses, we do think that the MR in combination with the network-adjusted observational analysis does provide some evidence that potential influence of genetic variation in the SPTLC3 gene region goes through Cer22:0 rather than other ceramides. (Either those were not strongly associated with SNPs in the SPTLC3 gene region, or the ceramides were not associated with T2D risk conditional on Cer22:0 levels.) However, because of the strong assumptions for the causal interpretation of MR-estimates we refrain from strongly emphasizing the MR-estimates for Cer22:0 as evidence for a 'causal effect'. In the revised version, we follow your advice and contextualize the MR-estimates with the other evidence that we generated.

9) To what extent do you think each of your analyses are robust to reverse causation, i.e. an effect of disease (sub-clinical in the case of the prospective association analyses) on ceramide levels?

Response: We have added sensitivity analysis excluding incident case that occurred within the first two years of follow-up (**Supplemental Table 8**). The results did not change considerably.

Minor points

10) Fasting effect – it looks from table 2b that there is an effect of fasting, at least on the total dh-ceramide levels, if not the total ceramide levels. I understand you fit fasting status as a covariate in your models, so its effect on individual analyses is not in question. However, I am wondering to what extent you think the patterns or association/correlation you see in these mostly fasted (>8 hours)

samples are indicative/representative of everyday function, i.e. non-fasted levels? For example, you talk in the discussion about ceramides concentrations acting as 'nutrient sensors' and being sensitive to 'metabolic challenge'.

Response: The absolute levels of total (dh)ceramides and ceramides are expected to vary with total blood lipid levels, and, therefore, with fasting status. Adjustment for fasting along with adjustment for total ceramide and dihydroceramide and cholesterol and triglyceride concentrations should control for the short-term effects of fasting status. The available evidence from dietary intervention trials indicates that the fatty acid composition of the diet affects the ceramide concentration over days and weeks (26). The current evidence, including our study, suggests that longer term composition of the habitual diet likely affects overall ceramide levels and might also influence the relative FA-composition within ceramides. More evidence from controlled dietary intervention studies is warranted to validate the effect of foods and food groups. Moreover, acute feeding trials that investigate the short-term dose-response curves of specific ceramides under dietary challenges would be very informative, but we are not aware of such studies.

11) Abstract - It would be useful to have an indication of the sample size (and case/control balance) at least for the main observational work in the Abstract

Response: Done.

12) Intro, p.2, L76 – "ceramides are among the most likely mediators of the effect of diet on cardiometabolic risk." – this sounds like there is only one set of mediators and that they are ceramides. More likely is that there are many mediators in this relationship.

Response: We meant to say that the available evidence qualifies ceramides as likely mediators – without claiming an exclusive role. We hope that comes across better after rephrasing the sentence: 'Due to their likely role as disease determinant and the demonstrated sensitivity to dietary exposures, ceramides are plausibly among metabolic mediators of the effect of diet on cardiometabolic risk.' (p3, L77-78)

13) Intro, p.2, L91-94 – This final sentence in the intro. is hard to read/comprehend.

Response: We hope the sentence reads easier after rephrasing: 'We also performed hypothesis-generating mediation analyses, estimating to what extent diet-related (dh)ceramide levels could explain the adverse effects of red meat consumption and the beneficial effects of coffee consumption on T2D risk.' (p3, L93-95)

14) Methods – there appears to be no adjustment for multiple testing (i.e. for testing 25 (dh)ceramides in Suppl. Table 3). Please comment on this.

Response: Thank you for this remark. The situation is complicated, because apart from testing several ceramides and dh-ceramides, we use multiple tests against H1 (to not reject H0). More specifically, we estimated the association between a lipid and the cardiometabolic endpoint in numerous models (depending on the number of direct neighbors 4 to 128 models per metabolite). If any of these models generated a non-significant p-value we do not reject H0, i.e. we did not classify the ceramide metabolite-cardiometabolic endpoint relation as direct effect.

15) Methods, P.23, L786 – you mention a "standardized procedure" – could you add further details about the time between sample collection, processing and freezing? And also, the temperature the sample was kept at in between these steps? Perhaps this has been published elsewhere?

Response: *Indeed, the study protocols including blood sampling procedures were described elsewhere. We extended the description of the blood sampling and cite the literature:*

'At baseline, blood samples were drawn under standardized conditions regarding room temperature according to the study protocol and stored in liquid nitrogen (-196°C) or deep freezers (-80°C). Per participant, 30 ml of blood were collected, of which 20 ml were filled in monovettes containing citrate. Samples were separated in serum, plasma, buffy coat, and erythrocytes and aliquoted into 0.5 ml straws as previously described in detail (90).' (p21, L545-549)

16) Methods, P.23-4 (Genetics) – did you check in the imputed data for any residual structure relating to the original genotyping batch (e.g. by running a GWAS for batch, and/or doing a PCA and looking for batch-related clustering)? If such structure existed and batch selections were based on, for example, case status for some disease, this could maybe create issues in the GWAS.

Response: *We now explain: 'For all laboratory measurements [including the GWAS], samples were randomly distributed across batches independent of case-status, and all laboratory and data-processing steps were performed blind to the case status.'* (p21, L551-552) *By design, any technical variance is completely at random (and thus unrelated to case-status) and no systematic errors are to be expected.*

17) Methods, P24, L863-4 – it's not clear to me what the values in brackets represent. I thought it was the imputation value but the values don't really make sense (e.g. 12 for BMI). Perhaps it is the number of missing values that had to be imputed? Please clarify.

Response: *True, there was information missing. You correctly suspected that the figures indicated the number of participants with missing values, which we now specify: '(waist circumference, $n_{\text{missing}}= 2$; BMI, $n_{\text{missing}}= 12$; blood lipids, $n_{\text{missing}}= 82$; blood pressure, $n_{\text{missing}}= 148$).'* (p23, 627-628)

Also, can you add more detail about the linear regression model used to determine the imputation values?

Response: *We added a more specific description of the imputation approach:*

'A moderate fraction of covariable information was missing (waist circumference, $n_{\text{missing}}= 2$; BMI, $n_{\text{missing}}= 12$; blood lipids, $n_{\text{missing}}= 82$; blood pressure, $n_{\text{missing}}= 148$). Single imputation was used to impute these missing values, applying the "predictive mean matching method" from the SAS procedure PROC MI. The "predictive mean matching method" draws information from other covariables to predict missing values and, compared with linear regression, generally generates more plausible imputed variable distributions (27). The following variables contributed to the prediction of missing values: incident case (T2D/CVD) during follow-up (yes, no), sex, age, height, smoking, leisure time physical activity (sports, biking, gardening), drug treatment (antihypertensive, lipid-lowering, aspirin), prevalent disease status (T2D, CVD), total energy intake, intakes of whole grain bread, grain flakes, grains, and muesli, fresh fruit, raw vegetables, cooked vegetables, nuts, coffee, high-energy soft drinks, fish, red meat and processed meat, total alcohol consumption, and educational attainment.' (p23, L626-636)

What about when categorical data were missing?

Response: *There were no missing values in categorical variables.*

18) Methods, P24, L864-5 – how was fasting status coded – is it a continuous trait (e.g. hours) or binary (categorisation based on more or less than 8 hours)?

Response: *The latter. We now specify: 'Fasting status was modeled as binary variable (≥ 8 hours, yes/no).'* (p23, L644)

19) Methods, P.25, L866 – what are the corresponding alcohol intake levels for the different categories, i.e. how are they defined?

Response: *We added the following definition in the Methods section: 'Alcohol intake was modeled in 6 sex-specific intake level categories. The categories in men were: abstainers, 0-6g/d, 6-12 g/d, 12-24 g/d, 24-60 g/d, 60-96 g/d, >96 g/d; The categories in women were: abstainers, 0-6g/d, 6-12 g/d, 12-24 g/d, 24-60 g/d, >60 g/d.'* (p23, L638-641)

20) Methods, P.26 (GWAS) – Plink v1.07 is listed in the software section but it's not clear where in the analysis it's used.

Response: *Thank you very much! We did not use PLINK, it was mistakenly included and is deleted from the revised software list.*

21) Methods, P.26 (GWAS) – please give details of the specific SNPTTEST model used. Also, there is no mention of the X chromosome – was data not available for this?

Response: *We had specified the genetic model just after citing SNP-test, but we added additional details: 'We assumed a frequentist additive genetic model (method expected: genotype dosage), adjusted for age at recruitment and sex.'* (p24, L694-696)

Data on the x-chromosome were indeed not available, and we have stated that in the Methods: 'Data were available for the 22 autosomes, but not for the sex chromosomes.' (p22, L596-597)

22) Methods, P.26 (GWAS) – is the GWAS also done on log-transformed z-scored data? Please clarify in the text.

Response: *Yes. We specified the modeling of the phenotype in the revised methods: 'Then, we used SNPtest v2.5.2 for exploratory single variant association analysis ($n \sim 5.339.213$ markers) as exposures and the log-transformed and z-standardized (dh)ceramides as an outcome.'* (p24, L691-693)

23) Results – a useful addition might be a correlation heatmap to show the basic correlation structure. This is a figure that will be familiar to most people and may help to demonstrate the extra information captured by the proposed NetCoupler-algorithm.

Response: Done (Supplementary Figure 2).

24) Results – for ease of reading, please include sample and ceramide numbers throughout the results (not just in the methods). For example, in the first section, state the no. of (dh)ceramides quantified and the no. of participants (including by case/control status).

Response: *Done. For example: 'The observational analyses were based on the measurement of 12 ceramides and 13 (dh)ceramides from a large lipidomics dataset in two case-cohort samples nested within the prospective EPIC-Potsdam study (775 participants with incident T2D among 1886 at-risk participants, and 551 participants with incident CVD among 1707 at-risk participants).'* (p4, L106-109)

25) Results, P.6, L177-8 – as I understand it, the GWAS was only run for the ten (dh)ceramides listed in table 1 – please clarify this (if true) in the text.

Response: *True. We are more specific about this now:*

'Using all participants in the representative EPIC-Potsdam subcohort with genetic and lipidomics data (n=1131), we conducted a GWAS with the plasma concentrations of the ten disease-related (dh)ceramides as the phenotypes.' (p8, L201-203)

26) Results, P.9, L230-2 – it's not clear to me what you mean by "GWAS signals below the genome-wide significant threshold". Did you use the full results for this, or do you mean those with $P < 10^{-5}$ (i.e. your suggestive significance level)?

Response: The gene set enrichment analysis uses the full GWAS results.

27) Results – Figure 3 – the quality of this figure needs improving as I can't make out the gene names on the Manhattan plots. Also, the legend sentence that begins "Shown are pathways .." doesn't make sense to me.

Response: We have substituted Figure 3 with a more comprehensive Table that summarizes the main findings from the GWAS, including effect estimates from other studies. The previous Figure 3 has been moved to the supplement, along with detailed Tables each of the ten ceramides, including comprehensive information for all ceramide associated SNPs with a p-value below $1E-05$.

28) Results (mediation) – the PE's seem high. Is this expected? To what extent could this be capturing effects of correlated ceramides.

Response: The question on the expected PE is difficult to answer. In general, we have selected the foods (coffee and red meat) because for both of them, lipid metabolism is a plausible mediator. Moreover, we selected the potential mediators according to association with the dietary exposures conditional on the other disease-related (dh)ceramides. Therefore, it is not implausible to observe a marked attenuation of the effect of the selected foods on T2D risk in the mediator-adjusted models, and other ceramides should not explain this effect attenuation.

29) It is not clear to me why the MR was only done for one of the SNP/ceramide pairs in Table 2. Is it because you wanted more than one instrument? Or because these SNPs weren't in the disease GWAS summary stats you were using? Further clarity is required on this point.

Response: Correct, we only performed MR on ceramides with several instruments that were genome-wide significant. The external validation in other cohorts that was included in the revised manuscript further supported that the effect of SNPs in the SPLTLC3-region on Cer22:0 was the most significant and consistent genetic signal. The level of confidence for other gene-ceramide associations was lower and we did therefor not use them in MR analyses.

Typos, grammar, etc.

30) Intro, p.2, L56/57 – "and thereby base prevention efforts" – this sentence doesn't make complete sense.

Response: We have rephrased the sentence:

'Pathway specific biomarkers can help to identify at-risk individuals and to discover the molecular processes that exposes them to higher cardiometabolic risk. Such biomarkers may also help to understand the influence of lifestyle on disease risk, enabling precise disease prevention.' (p3, L57-59)

31) Intro, p.2, L83 – "encoded in metabolomics network" – grammar issue here

Response: We have rephrased the sentence:

'Through adjusting for metabolomics network neighbors, our new *NetCoupler-algorithm* controls for confounding by biologically closely related metabolites. Thereby, the robust associations indicate direct effects of molecular markers on disease risk and are not attributable to the correlations with other metabolites.' (p3, L84-88)

Reviewer #4 (Remarks to the Author):

This is an interesting paper that throws several different causal inference techniques at the question of whether ceramides causally effect risk of T2D and CVD. This is a difficult – perhaps impossible – question to definitively answer with observational data. The authors make a noble attempt, and their results, together with experimental data from animals, are somewhat convincing – although I'm not ready to start using ceramides as surrogate markers for T2D or to believe that intervening on ceramides will decrease risk of T2D.

Response: Thank you for your interest. In the revised version, we have toned down the interpretation of our results, and consequently emphasize the hypothesis-generating nature of our work. For example, the concluding paragraph of the discussion was adjusted, removing the term 'surrogate marker':

'To conclude, our study indicates that the cardiometabolic risk associated with (dh)ceramide plasma concentrations depends on the contained acyl chain, especially if models are conditioned on other disease-related (dh)ceramides. These observations are consistent with the hypothesis that specific (dh)ceramides are involved in distinct molecular mechanisms of cardiometabolic disease etiology, which coincides with evidence from animal models. Our genetic analyses also suggested the implication of the disease-related (dh)ceramides in cardiometabolic disease-related molecular pathways. Furthermore, we showed that adjustment for a few T2D-related (dh)ceramides markedly attenuated the adverse effect of red meat and the protective effect of coffee consumption on T2D risk, consistent with the hypothesis that their effect on ceramide metabolism partially mediates the effect of these foods on T2D risk. Altogether, these results indicate that circulating (dh)ceramide profiles integrate information on the exposure to genetic and environmental cardiometabolic risk factors and may be applied as pathway-specific biomarkers for cardiometabolic health.' (p19, L460-471)

A few specific comments are below:

1. Comments about the NetCoupler-algorithm:

The NetCoupler-algorithm seems to be an empirical way of creating a network of ceramides based on their conditional correlations, and then investigating the direct effect of a specific ceramide on the outcome of interest (e.g., T2D), by seeing if its association with the outcome remains after fitting separate models adjusting for all possible combinations of the other adjacent variables. The thought is that if the ceramide has a direct effect on the outcome, then it will remain associated with the outcome whether one adjusts for other ceramides on the causal pathway and/or other ceramides that could be potentially confounding the relationship.

Response: Thank you for the precise summary. We have just one small remark. Adjusting for downstream ceramides on the causal path (mediators) would block the path and explain the effect of upstream causes on the endpoint. Only the downstream effectors, that is the ceramides that are closest to the molecular trigger of, for example, T2D pathogenesis are expected to remain associated.

1a. The NetCoupler algorithm appears that it may be sensitive to the order in which the ceramides are considered (because those found to have direct effects are included as potential confounders in subsequent models). How robust are the selected ceramides to the order in which they were considered?

Response: The network estimation is not order dependent. The identification of direct effects in a single

round of the NetCoupler-algorithm is also not order dependent because we iterate through all potential models. However, the empirical information from the different network-adjusted models can be ambiguous. If the direct effect is of interest, models must be adjusted for confounders and mediators. But adjustment for colliders must be avoided because it introduces spurious association. Therefore, it is not possible to specify the correct adjustment set if the models are not consistent without knowledge about effect directions. But it can be shown that variables with a direct effect can be confounders or mediators (which should be adjusted for) but not colliders. Therefore, we include the variables with an unambiguous direct effect in the fixed adjustment set and repeat the multi-model procedure. This may resolve the ambiguity for some of the previously ambiguous variables, possibly leading to the identification of additional direct effects. This process is repeated until no additional direct effects are identified. In other words, a single round of the NetCoupler algorithm is not order dependent, but the structural information from one round can help with identification of additional direct effects.

1b. Were Cox models properly weighted to account for the over-sampling of cases? This is not clear.

Response: *The description of the proportional hazards models was indeed insufficient, and we have updated the text:* Association between ceramides and disease risk was evaluated in Cox proportional hazards regression models with age as underlying time scale. Study exit was determined by diagnosis of diabetes, dropout, or censoring time, whichever came first. The case-cohort design was accounted for by Prentice weighting (105)' (p24, L669-672)

1c. Direct effects seem to be of interest, but total effects might be of more interest. If one ceramide leads to a cascading effect of other ceramides that then leads to T2D, you would want to capture the total effect of that ceramide. If the total effect is only through the cascade (indirect effect), then one would miss this ceramide, even though it is causing T2D.

Response: *Several lines of evidence demonstrate that ceramides are associated with cardiometabolic health. Evidence from animal models indicates that ceramides, depending on the contained acyl-chain, are involved in different T2D-related pathways. Therefore, it is important to understand the role of specific (dh)ceramides as determinants of human cardiometabolic health. To use your words, our network-adjustment strategy is expected to detect the ceramides at the end of the cascade, the ones that are closest to the endpoint of interest. The identification of (dh)ceramides for which the endpoint-associations are robust against network-adjustments may help to elucidate the biological mechanisms that implicate ceramides in cardiometabolic disease etiology. We also clearly acknowledge the hypothesis-generating nature of our study, emphasizing that it may help to integrate observed risk association with evidence from other study types.*

We clarify in the revised limitation section: 'In combination with experimental data, our selection of specific chain-length (dh)ceramides with direct effects on disease risk can be useful to elucidate molecular mechanisms. However, for other applications, including risk prediction, the total effect of a biomarker is more important, and it might not be advantageous to adjust for other correlated (dh)ceramides or total levels of ceramides and dihydroceramides.' (p18, L441-443)

1d. Removing Cer26:1 because it was non-significant in the joint model is somewhat unappealing and sort of runs counter to the authors' rationale for using the NetCoupler algorithm in the first place. Its non-significance must imply that one (or some) of the other ceramides are confounders or mediators

between its relationship with T2D. But that was ruled out in the original NetCoupler selection process, right? Non-significance is a crude measure of the importance of a variable.

***Response:** From a theoretical causal inference perspective, this is an example for the difference between identification and estimation. Theoretically, our network-based adjustment strategy should block all confounding and mediating paths. However, the ceramide network is estimated from empirical data distributions, and the selection of direct effects must rely on some sort of cutoff. We now explicitly acknowledge this disagreement between theory and empirical results. ('We also first classified one ceramide (Cer26:1) as having direct effect on T2D, but the significant association was rendered non-significant in the final T2D model that included all selected (dh)ceramides. This shows some disagreement between theoretical effect identification and empirical effect estimation, but all other selected (dh)ceramides remained statistically significantly associated in the final disease risk models.') (p18, L435-439)*

2. There seems to be a lot of reliance throughout the manuscript on significant p-values.

***Response:** Yes, we use p-value-based selection cutoffs. We had discussed this in the limitations, and in the revised version we acknowledge the limitation of p-value-based selection criteria even more clearly. ('The p-value-based variable selection and inferences in our observational and genetic analyses also depended on the sample size, which complicates comparison to studies with different statistical power.') (p18, L443-445) Concerning future developments of the NetCoupler approach, it would be interesting to implement machine learning-based variable selection methods that evaluate variable importance and robustness in population subsets rather than statistical significance. However, a comparison between different modeling approaches is beyond the scope of this study.*

Given the size of the dataset and the data preparation (log-transformation, removal of outliers), the inference about the most important variables based on parametric tests should be valid, and the results from non-parametric approaches should generally be comparable.

3. Comments about Mendelian randomization:

3a. Based on Mendelian randomization, Cer22:0 is said to be causally associated with T2D (estimate = 0.068, p=0.031) but not with CAD (estimate = -0.052, p=0.20). P-values are dependent on the number of outcomes. There were more T2D outcomes (775) than CAD outcomes (283, I think). Perhaps with more outcomes / bigger sample size, Cer22:0 would have crossed the magical 0.05 p-value threshold. (Again, this is a problem with reliance on statistical significance to infer causality.) However, a counter-argument to what I just said was that the CAD and stroke estimates went in opposite directions, which might be difficult to explain if there truly is a causal effect.

***Response:** According to the two-sample design, the Mendelian randomization estimates for the SNP endpoint-associations in our study were drawn from large GWAS-consortia (DIAGRAM, CARDioGRAM, MEGASTROKE). Statistical power should not be a major issue for the SNP-endpoint association.*

Beyond that and as mentioned above, we understand and partially share your reservations against p-value based decisions. However, particularly for the interpretation of MR results reliance on the statistical significance of the IV-estimate is the widely accepted standard. Deviation from that standard would bring about other issues, for example, in terms of comparability with other studies. In the revised limitation section, we explicitly discuss that the p-value-based selection of instruments for the MRs makes our study results dependent on sample size, implying the possibility that larger GWAS on (dh)ceramides may identify more IVs for additional MRs. ('The p-value-based variable selection and inferences in our observational

and genetic analyses also depended on the sample size, which complicates comparison to studies with different statistical power. Particularly, the GWAS on ceramide risk markers had limited statistical power. (p18, L443-446) [...] Our findings encourage larger GWAS on ceramides that may also generate more genetic instruments for MR studies.' (p19, L457-458)

3b. For what it's worth, I am skeptical of almost all Mendelian randomization studies. I have a hard time believing that all of the assumptions are met – particularly that the direct effect of the SNP on the outcome is only through its effect on the ceramide. VanderWeele et al., (2014) have a nice paper on some of the challenges of using this approach to infer causality (<https://pubmed.ncbi.nlm.nih.gov/24681576/>).

Response: We understand that several aspects of the interpretation of Mendelian Randomization studies are subject to controversial discussion. Specifically, the analogy with RCTs and the comparison of quantitative effect estimates from Mendelian Randomization studies with effect estimates from interventions or observational studies has been criticized by Tyler VanderWeele and, more recently, by Miguel Hernán and others.

However, these discussions have not discredited the Mendelian Randomization approach in general. The applications are numerous, and the use is increasing (see Figure below). Applications in fields where intervention studies are not amendable are backed by the general understanding that Mendelian Randomization studies are not sensitive to some sources of bias that can cause major issues with observational etiological studies, such as confounding and reverse causation. We agree that this should not result in any 'study hierarchy' and we carefully revised our text to not single out the MR results compared with our other findings. Also, we are now more careful with the causal interpretation of MR estimates. The Mendelian Randomization analyses are one of several epidemiological methods (a well-established one) that we apply to integrate genetic and observational evidence on the potential role of (dh)ceramides as determinants of cardiometabolic health.

Search query: mendelian randomization[title] OR mendelian randomisation[title]

4. Comments on Discussion:

4a. Consistency with animal results is reassuring.

Response: For us, too.

4b. Use as surrogate endpoints is a little premature/ over-interpreting results, especially given the relatively small HR for most of the ceramides.

Response: We have rephrased: ' Altogether, these results indicate that circulating (dh)ceramide profiles integrate information on the exposure to genetic and environmental cardiometabolic risk factors and may be applied as pathway-specific biomarkers for cardiometabolic health.' (p19, L469-471)

4c. I appreciated the authors discussion of their limitations – most of the limitations are things that I had identified while going through the manuscript (and are written perhaps a little more strongly above), and I agree with the limitations.

Response: Thank you. We take this as a great compliment.

5. Style: I'm not a regular reader of Nature Communication, so perhaps some of these critiques are due to the way that papers for this journal are often formatted, but I found myself flipping back and forth

frequently between Results and Methods. My understanding is that essential information is supposed to be included in Results and more technical information is included in Methods. However, I felt like essential information such as sample sizes of the cohorts, the timing of the measurement of ceramide concentrations, number of T2D and CVD events should have been included in the Results.

Response: Done.

6. Other comments:

6a. Line 749: Censoring date of 30st of November, 2006 does not exist.

Response: Thank you! Corrected.

6b. The density plots of the z-scores and the z-scores of the log-transformed variables (Figure 1A), are not very helpful for determining whether the normality assumption now holds. A better visualization is needed.

Response: We have improved the display of (dh)ceramide distributions (new Figure 1).

6c. "Comprehensive confounder adjustment" is a misnomer. You can never know if you have comprehensively adjusted for all confounders.

Response: Thank you for pointing this out. The 'misnomer' is quite common in the epidemiological literature, even among authors who are very critical about correct terminology and aware of the uncertainty about residual confounding in all observational analyses. We generally tend to interpret 'comprehensive' as 'extensive' rather than 'exhaustive'. We do agree, however, that room for misinterpretation should be avoided wherever possible. Now we write 'extensively confounder-adjusted' instead of 'comprehensively confounder-adjusted'.

1. Yoon S, Nguyen Hai C T, Yoo YJ, Kim J, Baik B, Kim S, Kim J, Kim S, Nam D. Efficient pathway enrichment and network analysis of GWAS summary data using GSA-SNP2. *Nucleic Acids Research*. 2018;46:e60-e60.
2. Hicks AA, Pramstaller PP, Johansson A, et al. Genetic determinants of circulating sphingolipid concentrations in European populations. *PLoS genetics*. 2009;5:e1000672.
3. Cresci S, Zhang R, Yang Q, Duncan MS, Xanthakis V, Jiang X, Vasan RS, Schaffer JE, Peterson LR. Genetic Architecture of Circulating Very-Long-Chain (C24:0 and C22:0) Ceramide Concentrations. *J Lipid Atheroscler*. 2020;9:172-183.
4. Wang DD, Toledo E, Hruby A, et al. Plasma Ceramides, Mediterranean Diet, and Incident Cardiovascular Disease in the PREDIMED Trial (Prevencion con Dieta Mediterranea). *Circulation*. 2017;135:2028-2040.
5. Van Dam RM, Hu FB. Coffee consumption and risk of type 2 diabetes: a systematic review. *Jama*. 2005;294:97-104.
6. Van Dam RM, Feskens EJ. Coffee consumption and risk of type 2 diabetes mellitus. *The Lancet*. 2002;360:1477-1478.

7. Salazar-Martinez E, Willett WC, Ascherio A, Manson JE, Leitzmann MF, Stampfer MJ, Hu FB. Coffee consumption and risk for type 2 diabetes mellitus. *Annals of internal medicine*. 2004;140:1-8.
8. Ding M, Bhupathiraju SN, Chen M, van Dam RM, Hu FB. Caffeinated and decaffeinated coffee consumption and risk of type 2 diabetes: a systematic review and a dose-response meta-analysis. *Diabetes care*. 2014;37:569-586.
9. Pan A, Sun Q, Bernstein AM, Schulze MB, Manson JE, Willett WC, Hu FB. Red meat consumption and risk of type 2 diabetes: 3 cohorts of US adults and an updated meta-analysis. *The American journal of clinical nutrition*. 2011;94:1088-1096.
10. Fung TT, Schulze M, Manson JE, Willett WC, Hu FB. Dietary patterns, meat intake, and the risk of type 2 diabetes in women. *Archives of internal medicine*. 2004;164:2235-2240.
11. Feskens EJ, Sluik D, van Woudenberg GJ. Meat consumption, diabetes, and its complications. *Current diabetes reports*. 2013;13:298-306.
12. Ohnaka K, Ikeda M, Maki T, Okada T, Shimazoe T, Adachi M, Nomura M, Takayanagi R, Kono S. Effects of 16-week consumption of caffeinated and decaffeinated instant coffee on glucose metabolism in a randomized controlled trial. *J Nutr Metab*. 2012;2012:207426.
13. MacKenzie T, Comi R, Sluss P, Keisari R, Manwar S, Kim J, Larson R, Baron JA. Metabolic and hormonal effects of caffeine: randomized, double-blind, placebo-controlled crossover trial. *Metabolism: clinical and experimental*. 2007;56:1694-1698.
14. van Dam RM, Pasma WJ, Verhoef P. Effects of coffee consumption on fasting blood glucose and insulin concentrations: randomized controlled trials in healthy volunteers. *Diabetes Care*. 2004;27:2990-2992.
15. van Nielen M, Feskens EJ, Rietman A, Siebelink E, Mensink M. Partly replacing meat protein with soy protein alters insulin resistance and blood lipids in postmenopausal women with abdominal obesity. *The Journal of nutrition*. 2014;144:1423-1429.
16. Navas-Carretero S, Pérez-Granados AM, Schoppen S, Vaquero MP. An oily fish diet increases insulin sensitivity compared to a red meat diet in young iron-deficient women. *Br J Nutr*. 2009;102:546-553.
17. Lontchi-Yimagou E, Sobngwi E, Matsha TE, Kengne AP. Diabetes mellitus and inflammation. *Current diabetes reports*. 2013;13:435-444.
18. Diedisheim M, Carcarino E, Vandiedonck C, Roussel R, Gautier J-F, Venteclef N. Regulation of inflammation in diabetes: From genetics to epigenomics evidence. *Molecular Metabolism*. 2020;41:101041.
19. Krumsiek J, Mittelstrass K, Do KT, et al. Gender-specific pathway differences in the human serum metabolome. *Metabolomics*. 2015;11:1815-1833.
20. VanderWeele TJ. Causal mediation analysis with survival data. *Epidemiology*. 2011;22:582-585.
21. Mahajan A, Taliun D, Thurner M, et al. Fine-mapping type 2 diabetes loci to single-variant resolution using high-density imputation and islet-specific epigenome maps. *Nat Genet*. 2018;50:1505-1513.
22. Nelson CP, Goel A, Butterworth AS, et al. Association analyses based on false discovery rate implicate new loci for coronary artery disease. *Nat Genet*. 2017;49:1385-1391.
23. Malik R, Chauhan G, Traylor M, et al. Multiancestry genome-wide association study of 520,000 subjects identifies 32 loci associated with stroke and stroke subtypes. *Nat Genet*. 2018;50:524-537.
24. Hemani G, Zheng J, Elsworth B, et al. The MR-Base platform supports systematic causal inference across the human phenome. *Elife*. 2018;7.
25. VanderWeele TJ, Tchetgen Tchetgen EJ, Cornelis M, Kraft P. Methodological challenges in mendelian randomization. *Epidemiology*. 2014;25:427-435.
26. Rosqvist F, Kullberg J, Ståhlman M, et al. Overeating Saturated Fat Promotes Fatty Liver and Ceramides Compared With Polyunsaturated Fat: A Randomized Trial. *J Clin Endocrinol Metab*. 2019;104:6207-6219.

27. Schenker N, Taylor JM. Partially parametric techniques for multiple imputation. *Computational statistics & data analysis*. 1996;22:425-446.

REVIEWER COMMENTS

Reviewer #1 (Remarks to the Author):

The assessment of ICC is interesting and relevant and I think this is a good addition to the manuscript. However, it does not address the issue of technical variance and should not be conflated with this. ICC does incorporate technical variance but also biological variance over time, so the discussion should reflect this. That was not my question. The technical variance still needs to be reported. If no data is available on this, then this should be recognized as a limitation and discussed. With regard to the baseline data for the cases. I appreciate that the use of appropriately weighted models adjusts can produce unbiased risk estimates. That was not the point of my request. Rather, it is simply for completion and to show if the cases did in deed have different characteristics from the baseline population. It should be a simple matter to include this information along with r explanation of how you dealt with this using the weighted models.

The authors have adequately addressed my other concerns.

Reviewer #2 (Remarks to the Author):

again, authors thanked for their very comprehensive changes to their paper in response to reviewer comments. However, I could not understand why you did not correct for multiple testing -m you gave a long answer but I was not fully convinced but I revert to stats reviewers here. I also could not work out whether the genetic analyses were repeated with exact same P values as appears in the table. But I sense the MR is not definitive - overall, whilst a complex field, I remain somewhat unconvinced about the findings as currently written. Also, are there any trial data to show coffee lowers diabetes risk? If not, please call this an association rather than an effect.

Reviewer #3 (Remarks to the Author):

This revised version is much improved, and I am satisfied with the responses to my previous queries, etc. I have just a few minor comments on the current version, as follows:

1. Intro – reference to ‘genome-wide association screens (GWAS)’ – GWAS conventionally used to refer to ‘genome-wide association studies’.
2. Results, p.5 – when you refer to ‘statistically significantly associated’, please add your criteria for this (FDR <0.05 I think). Also, it would be my preference to drop the ‘significantly’ altogether (see PMID: 11159626), but I understand this is a matter of opinion.
3. Results, p.5 – regarding the results presented in para. 1 of the section ‘Direct links between the (dh)ceramide network and cardiometabolic risk’ (and in Supp. Table 3), it’s not clear to me where the methods corresponding to this are. As far as I can tell these are the results of a Cox regression fitted for each (dh)ceramide in turn, i.e. not related to the network analysis? Or does this represent the first stage of the network analysis? Some clarity on this would be helpful.
4. Results – where Tables contain results from subset analyses (e.g. Supp. Tables 7 & 8) it would be useful to have the N stated in the legend.
5. Results, Table 3 – what are the units of the effect estimates? Are they the same across all studies?
6. Methods – for me, the description of the EPIC-Potsdam cohort (or at least the data used for this study) is still a little confusing. If I have understood it correctly, I think the statement that is potentially missing is the one that says that the subcohort minus those with a diagnosis for your diseases of interest made up the control groups. You should include in the methods the control N for each analysis (T2D/CVD) (explicitly, not expecting the reader to do the sums). Perhaps a reference that describes the nested case-cohort approach would be helpful for those less familiar with this study design?

7. Methods – I suggest moving the description of the ‘Summary-level data from GWAS-Consortia’ further down the methods towards the GWAS/MR section.

Reviewer #4 (Remarks to the Author):

The authors were quite responsive to my previous concerns as well as those of the other reviewers. Although not all of my concerns have gone away (I am still skeptical of Mendelian randomization and I still think this paper relies a lot on p-values), I believe the authors understand the issues, their choices are arguably reasonable, and they clearly highlight limitations in the manuscript. I also want to point out that I agree with the authors' justification for not adjusting for multiple comparisons. So I am OK with the revised manuscript. Just one final point to the authors:

Mendelian randomization is clearly becoming more and more popular, as illustrated with your figure. However, you could draw a similar figure showing the exponential increased use of statistical significance as $p < 0.05$ in the medical literature, yet it is now pretty well recognized that science is overly-reliant on “statistical significance.” As I am sure you recognize, popularity of a procedure does not imply that it is a sound procedure.

REVIEWER COMMENTS

IMPORTANT GENERAL REMARK: We were informed by the Human Study Center at the German Institute of Human Nutrition that a coding error occurred regarding the case and subcohort status of several participants in the EPIC-Potsdam lipidomics dataset. Among other projects, the datasets underlying the present analysis were affected. We needed to rerun all analyses with the updated datasets. The results have changed, including the selected T2D and CVD risk markers, and Tables, Figures, and text were adjusted accordingly. We apologize for the necessary corrections at this advanced stage. However, we would like to emphasize, despite the upstream error in the data-management, the applied workflow and methods remained unchanged.

Formats:

- Original reviewer comments: **bold**
- Our responses to the comments: **blue**
- Changed passage from the revised manuscript: *italics*
- References to the clean, revised manuscript: **(Lines X-Y)**

Reviewer #1 (Remarks to the Author):

The assessment of ICC is interesting and relevant and I think this is a good addition to the manuscript. However, it does not address the issue of technical variance and should not be conflated with this. ICC does incorporate technical variance but also biological variance over time, so the discussion should reflect this. That was not my question. The technical variance still needs to be reported. If no data is available on this, then this should be recognized as a limitation and discussed.

We have additionally included the (quite general) information about technical validity that Metabolon provided concerning the global lipidomics panel.

According to Metabolon®, the coefficients of variation (CVs) of lipid class concentrations are all below 10% and the median CV of species at a 1uM concentration in serum or plasma is approximately 5%. (Lines 516-518)

We would have wished for more transparency from the data-generating company. Now, we address this issue in the limitations section of the discussion.

Although delivering a very comprehensive lipidomics screen, the manufacturer (Metabolon®) did not disclose indicators of the technical variance for single lipids. We partly compensated for this lack of transparency by assessing the intra-individual variance (ICCs) of the single lipid measurements over several weeks in a pilot study, implying that we could not differentiate between technical and biological variance. (Lines 408-412)

With regard to the baseline data for the cases. I appreciate that the use of appropriately weighted models adjusts can produce unbiased risk estimates. That was not the point of my request. Rather, it is simply for completion and to show if the cases did in deed have different characteristics from the baseline population. It should be a simple matter to include this information along with r explanation of how you dealt with this using the weighted models.

Sorry for not getting that point right in the first revision. The presentation of a baseline Table with confounder distribution across exposure categories is an essential component of a case-cohort study

(Suppl. Tables 2a and 2b). Indeed, it is a simple matter also to compare confounder distributions between participants with and without incident type 2 diabetes (**new Suppl. Tables 2c and 2d**).

The authors have adequately addressed my other concerns.

We are very grateful for your valuable and much-appreciated contributions!

Reviewer #2 (Remarks to the Author):

again, authors thanked for their very comprehensive changes to their paper in response to reviewer comments.

First and foremost, we want to thank reviewer Reviewer #2 for the general approval of our edits, which depended on the very thoughtful and constructive criticism we received with the first revision!

However, I could not understand why you did not correct for multiple testing -m you gave a long answer but I was not fully convinced but I revert to stats reviewers here.

We have evaluated the robustness of estimates across differently adjusted models. Reviewer #4 explicitly agrees with our standpoint that multiple-testing correction is not indicated in this workflow. However, we also provide the multiple testing corrected type 2 diabetes hazard ratios of all ceramides from differently adjusted Cox models (**Supplemental Table 3**). Therefore, the reader has access to comprehensive data to compare our selection and risk estimation procedure with the standard approach of multiple-testing corrected single metabolite estimates.

I also could not work out whether the genetic analyses were repeated with exact same P values as appears in the table. But I sense the MR is not definitive - overall, whilst a complex field, I remain somewhat unconvinced about the findings as currently written.

For the MR analyses, we used the estimates for the SNP-(dh) ceramide associations from three GWAS in different source populations. Hence, the input estimates and standard errors of the SNP-exposure association in the MR differed. The estimates and standard error of the SNP-outcome association in all MR were from DIAGRAM. In the revised version, we further caution not to over-interpret the MR results: *However, the MR estimates alone do not provide conclusive evidence on causality but complement the observational estimates due to distinct sources of bias.* (**Lines 434-435**)

Also, are there any trial data to show coffee lowers diabetes risk? If not, please call this an association rather than an effect.

One goal of our analyses was to complement the extensive observational evidence on the link between coffee consumption and type 2 diabetes risk by generating data-supported hypotheses on possible mediators of an assumed biological effect of coffee on diabetes risk. An essential advantage of identifying potential intermediary factors is that they can help integrate epidemiological studies that follow their participants over decades with shorter intervention trials assessing changes in risk factor profiles in response to controlled coffee exposure. Hence we applied causal inference techniques and, therefore, cannot avoid causal language. (A mediation analysis only makes sense when assuming an effect of the exposure on the outcome.) However, we explicitly acknowledge the hypothesis-generating

nature of the mediation analyses [For example, by stating: 'Mediation analyses in observational data do not prove causality but generate testable hypotheses, which warrant validation in controlled trials' (Lines 436-438)]. We have also further revised the text and carefully include adjectives that temper all causal language. For example, we have written that we intended to find *potential* mediators (e.g., Lines 698 and 708) of a *putative* effect (e.g., Lines 49, 271, 286, 290/291) of coffee consumption on T2D risk.

Reviewer #3 (Remarks to the Author):

This revised version is much improved, and I am satisfied with the responses to my previous queries, etc. I have just a few minor comments on the current version, as follows:

Thank you! We greatly appreciate your original critiques and the positive feedback on the revised version.

1. Intro – reference to 'genome-wide association screens (GWAS)' – GWAS conventionally used to refer to 'genome-wide association studies'.

We adjusted the text and used the standard expansion of the abbreviation.

'genome-wide association studies (GWAS)' (Line 94)

2. Results, p.5 – when you refer to 'statistically significantly associated', please add your criteria for this (FDR <0.05 I think). Also, it would be my preference to drop the 'significantly' altogether (see PMID: 11159626), but I understand this is a matter of opinion.

We hope that our paper addresses some of the issues outlined in the referenced piece by Sterne and Davey Smith, though certainly not all. We do not simply report significant findings but look for consistency across differently adjusted models, using subsets of correlated (dh)ceramides, and we examine the consistency of risk estimates with other methods (MR and gene set enrichment). We admit that reliance on p-values introduces arbitrary cutoffs. Therefore, our discussion acknowledges limitations related to statistical significance-based marker selection.

We have added the criteria that define statistical significance wherever applicable in the result section, e.g. (FDR<0.05) (Line 94).

3. Results, p.5 – regarding the results presented in para. 1 of the section 'Direct links between the (dh)ceramide network and cardiometabolic risk' (and in Supp. Table 3), it's not clear to me where the methods corresponding to this are. As far as I can tell these are the results of a Cox regression fitted for each (dh)ceramide in turn, i.e. not related to the network analysis? Or does this represent the first stage of the network analysis? Some clarity on this would be helpful.

The first section indeed presents risk associations from non-network-adjusted, single metabolite Cox models. Thanks for pointing this lack of clarity out. We have included an *additional sub-header* (Line 132/133) and integrated more specific descriptions of the workflow in the results section.

4. Results – where Tables contain results from subset analyses (e.g. Supp. Tables 7 & 8) it would be useful to have the N stated in the legend.

We have included the critical sample size information in supplemental tables 7 and 8.

5. Results, Table 3 – what are the units of the effect estimates? Are they the same across all studies?

The units of the MR estimates rely on units that were used in the respective exposure-GWAS. Hence, the MR estimates for risk of type 2 diabetes correspond to an per SD-increase of log-transformed Cer22:0 in EPIC-Potsdam and EUROSPAN and to a per unit (in $\mu\text{g/ml}$) increase of Cer22:0 in FHSOC. We have included this information in Table 3's footnote.

6. Methods – for me, the description of the EPIC-Potsdam cohort (or at least the data used for this study) is still a little confusing. If I have understood it correctly, I think the statement that is potentially missing is the one that says that the subcohort minus those with a diagnosis for your diseases of interest made up the control groups. You should include in the methods the control N for each analysis (T2D/CVD) (explicitly, not expecting the reader to do the sums). Perhaps a reference that describes the nested case-cohort approach would be helpful for those less familiar with this study design?

Though statistically highly efficient, the case-cohort design can indeed be somewhat confusing. The intuitive cases vs. controls distinction from the standard case-control design is not directly applicable. The exposure and confounder distributions of all cases are compared to the subcohort. The subcohort represents the full cohort and, therefore, randomly includes some cases. The oversampling of cases is accounted for by weighted modeling. Hence, the modeling approach distinguishes between non-cases (members of the subcohort without incident disease), internal cases (members of the subcohort with incident disease), and external cases (all incident cases in the full cohort that were not randomly included in the subcohort).

We further revised the description of the case-cohort design and included an additional reference.

The case-cohort design relies on a randomly drawn subsample (the subcohort) and oversampling of all incident disease cases in the full cohort during the study period to boost the statistical power. Statistically accounting for the oversampling of cases, this design provides unbiased risk estimates for the full cohort (77). (Lines 463-466)

77. Wacholder S, Boivin JF. External comparisons with the case-cohort design. *Am J Epidemiol.* 1987;126:1198-1209.

7. Methods – I suggest moving the description of the 'Summary-level data from GWAS-Consortia' further down the methods towards the GWAS/MR section.

Done. (Now lines 670-675)

Reviewer #4 (Remarks to the Author):

The authors were quite responsive to my previous concerns as well as those of the other reviewers. Although not all of my concerns have gone away (I am still skeptical of Mendelian randomization and I still think this paper relies a lot on p-values), I believe the authors understand the issues, their choices are arguably reasonable, and they clearly highlight limitations in the manuscript. I also want to point out that I agree with the authors' justification for not adjusting for multiple comparisons. So I am OK with the revised manuscript.

Thank you for your thoughtful and instructive comments. We have further reduced reliance on p-values for the GWAS results, emphasizing the external validity of the results.

Just one final point to the authors:

Mendelian randomization is clearly becoming more and more popular, as illustrated with your figure. However, you could draw a similar figure showing the exponential increased use of statistical significance as $p < 0.05$ in the medical literature, yet it is now pretty well recognized that science is overly-reliant on "statistical significance." As I am sure you recognize, popularity of a procedure does not imply that it is a sound procedure.

The popularity of a method is indeed not the best indicator of its scientific value. Thus, a diagram of published studies was not the best way to address your valid concerns. Still, we think that the distinct sources of bias for observational and genetic proxy analyses justify their joint application. We fully agree with not equating MR with a causal interpretation of the estimates, which can be challenging with the established vocabulary. We hope the temperate presentation and interpretation of the genetic-proxy analysis in the revised manuscript version is acceptable. As mentioned above, we state: *However, the MR estimates alone do not provide conclusive evidence on causality but complement the observational estimates due to distinct sources of bias.* **(Lines 434-435)**

REVIEWER COMMENTS

Reviewer #1 (Remarks to the Author):

The authors have adequately addressed my concerns

Reviewer #3 (Remarks to the Author):

Considering the update made to the datasets underpinning this work, I opted to review the paper in its entirety once again. Given this and the time that has elapsed since my initial review of this manuscript, perhaps inevitably I have identified some minor areas for improvement that I missed previously. These are listed below, and I would encourage the authors to consider them.

1. Results, p.6, L106-8 – state the ICC coefficient you are using to define ‘fair to excellent reliability’ versus ‘poor reliability’. Also, is this reliability or repeatability? The measurements might be accurate but there could be a genuine lack of temporal stability. Same point re. reference to ‘reliability’ in methods.
2. Results, supp. Fig. 2 – state in legend which are partial correlations (i.e., upper triangle)
3. Results, Figure 3 – I understand that to keep the figure uncluttered you only have betas in parts A & C, but I think these results need to be in full somewhere (i.e., with corresponding 95% CI and p-values).
4. Discussion – re. the sentence that has been added re. the temporal stability of measures - consider modifying in response to my comment above and also because the text from ‘implying that we could not differentiate’ doesn’t make sense to me.
5. Discussion – the limitations paragraph is a bit ‘clunky’ – whilst I recognize this is largely because of responding to reviewer comments, you might consider integrating limitations into the relevant section of the discussion above, e.g., MR limitations at end of MR results discussion to aid readability (unless journal requirements specify you need a separate paragraph).
6. Methods – is there any sample overlap in the cohorts used for the two-sample MR – that is did any of the cohorts in the SNP-ceramide GWAS also contribute data to DIAGRAM? It would be useful add a statement regarding this point.
7. Methods – P36, L703 – not sure what ‘or anthropometric trait’ is referring to – I thought that coffee and red meat consumption were the only two exposures investigated as mediators?

Reviewer #3 (Remarks to the Author):

Considering the update made to the datasets underpinning this work, I opted to review the paper in its entirety once again. Given this and the time that has elapsed since my initial review of this manuscript, perhaps inevitably I have identified some minor areas for improvement that I missed previously. These are listed below, and I would encourage the authors to consider them.

We greatly appreciate the reviewer's time and exceptional commitment. Thank you very much!

1. Results, p.6, L106-8 – state the ICC coefficient you are using to define 'fair to excellent reliability' versus 'poor reliability'. Also, is this reliability or repeatability? The measurements might be accurate but there could be a genuine lack of temporal stability. Same point re. reference to 'reliability' in methods.

RESPONSE: As described in the methods, we performed a reliability study with blood samples collected several weeks apart and reported the ICC, an indicator of intraindividual temporal stability. We have extended the description in the methods section, including the definition of ICC classes according to Rosner.

"In a preceding analysis, we estimated intraclass correlation coefficients (ICCs) of repeated blood samples taken several weeks apart. The ICC relates intraindividual to between-person variation, indicating biological stability the measurements, and we used Rosner's classification of ICCs (ICC <0.40 poor reproducibility; ICC from 0.40–0.75 fair to good reproducibility; ICC >0.75 excellent reproducibility) (85)." (lines 457-461)

2. Results, supp. Fig. 2 – state in legend which are partial correlations (i.e., upper triangle)

RESPONSE: We have modified the legend of Supplementary Figure 2.

Supplementary Figure 2: Correlations (lower left triangle) and partial correlations (upper left triangle) among (dh)ceramides. Partial correlations are conditional on all other (dh)ceramides.

3. Results, Figure 3 – I understand that to keep the figure uncluttered you only have betas in parts A & C, but I think these results need to be in full somewhere (i.e., with corresponding 95% CI and p-values).

To keep the figure sparse, we have included more information in the legend of Figure 3: *Adjusted effect estimates (beta coefficients) of red meat on T2D-related (dh)ceramides (direction of associations consistent with mediation hypothesis; p-values < 0.05, one sided t-test).*

4. Discussion – re. the sentence that has been added re. the temporal stability of measures - consider modifying in response to my comment above and also because the text from 'implying that we could not differentiate' doesn't make sense to me.

RESPONSE: We have modified the sentence and replaced the highlighted part.

'We partly compensated for this lack of transparency by assessing the intra-individual variance of the single lipid measurements over several weeks in a pilot study, assessing the temporal stability of the (dh)ceramide measurements.' (lines 347-350)

5. Discussion – the limitations paragraph is a bit ‘clunky’ – whilst I recognize this is largely because of responding to reviewer comments, you might consider integrating limitations into the relevant section of the discussion above, e.g., MR limitations at end of MR results discussion to aid readability (unless journal requirements specify you need a separate paragraph).

RESPONSE: We agree that the paragraph was very long and have structured it into several smaller paragraphs. However, we would suggest keeping the comprehensive discussion of study limitations together. (lines 346-376)

6. Methods – is there any sample overlap in the cohorts used for the two-sample MR – that is did any of the cohorts in the SNP-ceramide GWAS also contribute data to DIAGRAM? It would be useful add a statement regarding this point.

RESPONSE: We added this information:

‘The EPIC-Potsdam GWAS data was included in the EPIC-Interact Consortium, which contributed GWAS data to the used DIAGRAM meta-analysis. The EUROSPAN and FHOCS cohorts did not contribute to the DIAGRAM data of the utilized publication (42).’ (lines 615-618)

7. Methods – P36, L703 – not sure what ‘or anthropometric trait’ is referring to – I thought that coffee and red meat consumption were the only two exposures investigated as mediators?

RESPONSE: Thank you! Deleted.